**Investigation**

# Shared evolutionary processes shape landscapes of genomic variation in the great apes

Murillo F. Rodrigues [1,2,*] Andrew D. Kern [1,2,†] Peter L. Ralph [1,2,3,†]

[1]Institute of Ecology and Evolution, University of Oregon, Eugene, OR 97403, USA
[2]Department of Biology, University of Oregon, Eugene, OR 97403, USA
[3]Department of Mathematics, University of Oregon, Eugene, OR 97403, USA

*Corresponding author: Department of Biology, University of Oregon, Eugene, OR 97403, USA. Email: murillor@uoregon.edu
[†]These authors contributed equally to this work.

For at least the past 5 decades, population genetics, as a field, has worked to describe the precise balance of forces that shape patterns of variation in genomes. The problem is challenging because modeling the interactions between evolutionary processes is difficult, and different processes can impact genetic variation in similar ways. In this paper, we describe how diversity and divergence between closely related species change with time, using correlations between landscapes of genetic variation as a tool to understand the interplay between evolutionary processes. We find strong correlations between landscapes of diversity and divergence in a well-sampled set of great ape genomes, and explore how various processes such as incomplete lineage sorting, mutation rate variation, GC-biased gene conversion and selection contribute to these correlations. Through highly realistic, chromosome-scale, forward-in-time simulations, we show that the landscapes of diversity and divergence in the great apes are too well correlated to be explained via strictly neutral processes alone. Our best fitting simulation includes both deleterious and beneficial mutations in functional portions of the genome, in which 9% of fixations within those regions is driven by positive selection. This study provides a framework for modeling genetic variation in closely related species, an approach which can shed light on the complex balance of forces that have shaped genetic variation.

Keywords: linked selection; selective sweep; background selection; GC-biased gene conversion; mutation rate variation; simulation

## Introduction

Genetic variation is determined by the combined action of mutation, demographic processes, recombination, and natural selection. However, there is still no consensus on the relative contributions of these processes and their interactions in shaping patterns of genetic variation. Two major open questions are: how does the influence of selection compare to other processes? And, to what degree is genetic variation influenced by beneficial vs deleterious mutations?

Genetic variation can be measured within a species or between species with 2 related metrics: within-species genetic diversity and between-species genetic divergence. Both can be estimated with genetic data by computing the per site average number of differences between pairs of samples within a species or between 2 species, and these are estimates of the mean time to coalescence. (Note that we do not discuss *relative divergence*, which is often measured using $F_{ST}$). Evolutionary processes impact diversity and divergence in different ways, so the relationship between these carries information regarding these processes.

Natural selection directly impacts genetic diversity because it can reduce the frequencies of alleles that are deleterious (negative selection) or increase those of beneficial alleles (positive selection). Selection can also directly affect between-species genetic divergence. Deleterious alleles are more likely to be lost from the population, thus reducing divergence at the affected ssites. On the other hand, beneficial alleles have a higher probability of fixation. This leads to an increase in the rate of substitution at sites under positive selection that in turn increases divergence between species. Thus, contrasting patterns of diversity and divergence at the same time can help disentangle between modes of selection (Hudson *et al.* 1987). Indeed, perhaps the most widely used test for detecting adaptive evolution, the McDonald–Kreitman test compares diversity and divergence contrasted between neutral (e.g. synonymous) and functional (e.g. nonsynonymous) site classes (McDonald and Kreitman 1991). This test and its extensions have been applied to a myriad of taxa, and it has become clear that a substantial proportion of amino acid substitutions are driven by positive selection in a number of taxa (Smith and Eyre-Walker 2002; Ingvarsson 2010; Slotte 2014; Galtier 2016).

Selection also disturbs genetic variation at nearby locations on the genome, and this indirect effect of selection on diversity is called "linked selection". Linked selection can be caused by at least 2 familiar mechanisms: genetic hitchhiking and background selection. Under genetic hitchhiking, as a beneficial mutation quickly increases in frequency in a population, its nearby genetic background is carried along, causing local reductions in levels of genetic diversity. The size of the region affected by the sweep depends on the strength of selection, which determines how fast fixation happens, and the crossover rate, because recombination allows linked sites to escape from the haplotype carrying the beneficial mutation (Smith and Haig 1974; Kaplan *et al.* 1989).

Under background selection, neutral variation linked to deleterious mutations is removed from the population unless, as before, focal lineages escape via recombination (Charlesworth *et al.* 1993). Both of these processes leave similar footprints on patterns of within-species genetic diversity, and so attempts to determine the contributions of positive and negative selection in shaping levels of genetic variation genomewide have proven to be difficult (Kim and Stephan 2000; Andolfatto 2001), although the processes seem separable more locally (Schrider and Kern 2017; Schrider 2020). Importantly, linked selection does not affect between-species genetic divergence as strongly, as a beneficial or deleterious mutation does not affect the substitution rate of linked, neutral mutations (Birky and Walsh 1988) (although it does affect divergence through ancestral levels of polymorphism Begun *et al.* 2007; Phung *et al.* 2016).

The effects of linked selection in shaping genetic variation are pervasive across genomes (Begun and Aquadro 1992; Cai *et al.* 2009; Lohmueller *et al.* 2011; Corbett-Detig *et al.* 2015; Murphy *et al.* 2022). For example, dips in nucleotide diversity surrounding functional substitutions have been uncovered in many taxa, such as fruit flies (Kern *et al.* 2002; Sattath *et al.* 2011), rodents (Halligan *et al.* 2013), *Capsella* (Williamson *et al.* 2014), and maize (Beissinger *et al.* 2016). In *Drosophila melanogaster*, levels of synonymous diversity (which is putatively neutral) and amino acid divergence are negatively correlated (Andolfatto 2007; Macpherson *et al.* 2007); positive selection can cause such a pattern if beneficial amino acid mutations are fixing and as they do reducing levels of linked neutral variation via selective sweeps. In contrast in humans, levels of synonymous diversity are roughly the same near amino acid substitutions and synonymous substitutions, suggesting recent, recent fixations at amino acids sites may not be the result of strongly beneficial alleles (Hernandez *et al.* 2011; Lohmueller *et al.* 2011). However, in the human genome, amino acid substitutions tend to be located in regions of lower constraint than synonymous substitutions, implying that the signal of positive selection may be confounded by the effects of background selection (Enard *et al.* 2014).

Two major challenges remain in the way of a fuller characterization of the effects of selection on genetic variation: (1) it is hard to model interactions between evolutionary processes (e.g. sweeps within highly constrained regions) and (2) model identifiability is challenging for some summaries of the data (e.g. sweeps and background selection may impact diversity in similar ways). Recent computational advances have made it possible for us to move from simpler backwards-in-time coalescent models (Hudson 1983) to more complex and computationally demanding forward-in-time simulations, and these have provided a route to studying these hard to model interactions between evolutionary processes across multiple sites (Kelleher *et al.* 2016; Haller and Messer 2019; Haller *et al.* 2019). Simulation-based inference can then allow us to better describe the roles of different modes of selection and other processes in shaping genomic variation. However, the problem of identifying features of the data that are informative of the strength and mode of selection still remains.

One promising approach might be to compare patterns of genetic variation in multiple species jointly as each species can be thought of as semi-independent realizations of the same evolutionary processes (c.f. Won and Hey 2005). In speciation genomics studies, it is common to visualize large-scale patterns of genetic variation along chromosomes (so-called landscapes of diversity and divergence), which may contain substantial information to help us disentangle evolutionary processes. Earlier empirical

surveys have focused on the identification of regions of accentuated relative divergence between populations (Turner *et al.* 2005; Harr 2006; Cruickshank and Hahn 2014), although patches of increased divergence can be the result of myriad forces besides reproductive isolation and adaptation. Recent comparative studies have found that landscapes of diversity are highly correlated between related groups of species, such as *Ficedula* flycatchers (Ellegren *et al.* 2012; Burri *et al.* 2015), warblers (Irwin *et al.* 2016), stonechats (van Doren *et al.* 2017), hummingbirds (Battey 2020), monkeyflowers (Stankowski *et al.* 2019) and *Populus* (Wang *et al.* 2020). Burri (2017) proposed that we could capitalize on correlated genomic landscapes to study the interplay between different forms of selection and other evolutionary processes. Neutral processes, such as retained ancestral diversity (i.e. incomplete lineage sorting, ILS) or migration, could potentially produce significant correlations in levels of diversity across species, however strong correlations have been observed among taxa with long divergence times and without evidence of gene flow. For example, Stankowski *et al.* (2019) found that landscapes of diversity and divergence are highly correlated across a radiation of monkeyflowers which spans 1 million year (or about $10N_e$ generations, where $N_e$ is the effective population size), far longer than the time scale on which we expect to see effects of ancestral variation and ILS (since the coalescent timescale spans just a few multiples of $N_e$). However, a shared process that independently occurs in the branches of a group of species could maintain correlations over long timescales. For example, if 2 species' physical arrangement of functional elements and local recombination rates are similar, the direct and indirect effects of selection could make it so that peaks and valleys on the landscape of diversity are similar, maintaining correlation between their landscapes over evolutionary time (Burri 2017; Delmore *et al.* 2018). Further, if mutational processes are heterogeneous across the genome in a manner that is shared among species, then correlated landscapes of diversity could be created through mutational variation as well.

Here, we aim to (1) describe whether and in what ways landscapes of within-species diversity and between-species divergence are correlated and (2) to tease apart the relative roles of positive and negative selection and other processes (e.g. ancestral variation, mutation rate variation) in shaping patterns of genetic variation. To understand processes driving these correlations, we employ highly realistic, chromosome-scale, forward-in-time simulations, since analytical predictions are not available. We use the great apes as a system to investigate correlated patterns of genetic variation because there is high quality population genomic data for all species (Prado-Martinez *et al.* 2013), the clade is about 12 million years old or $60N_e$ generations (but there have not been many chromosomal arrangements Jauch *et al.* 1992), and lastly the landscapes of gene density, recombination rate and mutation rate are roughly conserved (Stevison *et al.* 2016; Kronenberg *et al.* 2018). Our study demonstrates that correlated landscapes can be useful in distinguishing between modes of selection and the balance of direct and linked selection shaping genomic variation.

## Methods
### Genomic data
We retrieved single nucleotide polymorphism (SNP) calls for 10 great ape populations made on high coverage (~25×) short-read sequencing data from the Great Ape Genome Project (Prado-Martinez *et al.* 2013), mapped onto the human reference genome (NCBI36/hg18). We analyzed 86 individuals divided into

the following populations: human ($n = 9$ samples), bonobo ($n = 13$), Nigeria-Cameroon chimpanzee ($n = 10$), eastern chimpanzee ($n = 6$), central chimpanzee ($n = 4$), western chimpanzee ($n = 4$), eastern lowland gorilla ($n = 3$), western gorilla ($n = 27$), Sumatran orangutan ($n = 5$), Bornean orangutan ($n = 5$) (we excluded 2 samples from the original dataset: the Cross River gorilla and the chimpanzee hybrid). Prado-Martinez et al. (2013) applied several quality filters to the SNP calls (see Section 2.1 of their Supplementary Information) and, for each species, identified the genomic regions in which it would be unreliable to call SNPs (uncallable regions). For our downstream analyses, we only considered sites which were callable in all populations.

We calculated nucleotide diversity and divergence ($d_{XY}$) in non-overlapping 1 Mb windows using scikit-allel (Miles et al. 2020). Windows in which there were less than 40% callable sites were not used in any of the analyses. For example, this yielded 129 (out of 132) 1 Mb windows in chromosome 12 in which 75% of the sites were callable on average.

To tease apart the effects of GC-biased gene conversion (gBGC), we decomposed diversity and divergence by allelic states. gBGC is expected to affect weak bases (A or T) which are disfavored when in heterozygotes which also carry a strong base (G or C). Thus, one way understand the effects of gBGC is by comparing sites which were weak to those that were strong in the ancestor (ancestrally strong alleles are not affected by gBGC, but ancestrally weak alleles can be). We assumed that the state in the ancestor of the great apes to be the state seen in rhesus macaques (genome version RheMac2)—sites without enough information in RheMac2 were excluded. Then, we computed divergence only considering sites which were ancestrally weak or ancestrally strong (Supplementary Fig. 4). This approach has 2 major drawbacks: (1) many of the sites cannot be used because they are missing in RheMac2 and (2) sites can be mispolarized. Thus, we came up with a second approach to tease apart the effects of gBGC on correlations between genomic landscapes. When comparing 2 landscapes of divergence (which encompass 4 species), we can classify each site by the change in state that happened without needing to polarize mutations by looking at the ancestor. For example, if we have allelic states for 4 species and we see A–A–T–T as the configuration of alleles at a particular site, we know that there must have been 1 mutation which changed the state from a weak base to another weak base (W–W). On the other hand, if we see A–G–A–A there must have been 1 mutation from weak to strong (W–S) (or vice versa). Sites with multiple mutations (e.g. A–G–G–C) were removed from the analyses. Sites that did not change from W to S (or vice versa) are not expected to be affected by gBGC, and we refer to these as W–W or S–S mutations (Fig. 6a). Sites where there may have been a weak to strong change (W–S mutations) may be affected by gBGC (Fig. 6b). We only considered windows with at least 5% of callable sites in these analyses.

## Simulations

We implemented forward-in-time Wright–Fisher simulations of the entire evolutionary history of the great apes using SLiM (Haller and Messer 2019; Haller et al. 2019). Each branch in the great apes' tree was simulated as a single population with constant size (Fig. 1). Population splits occurred in a single generation, and there was no contact between populations postsplit. Population sizes and split times were taken from the estimates in Prado-Martinez et al. (2013). Across all our simulations, we simulated crossover events occurred with the sex-averaged rates from the deCODE genetic map (in assembly NCBI36/hg18

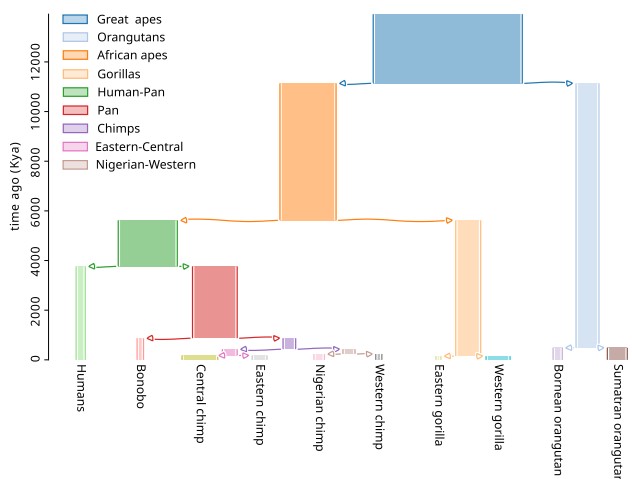

**Fig. 1.** Simulated great apes history. Arrows indicate population splits. Branch widths are proportional to population size. For example, the population size was 125, 089 for the great apes branch and 7, 672 for the humans branch. Figure was produced using *demesdraw* (Gower et al. 2022).

coordinates) (Kong et al. 2002). We then computed diversity and divergence in the same windows used for the real data using tskit (Kelleher et al. 2018; Ralph et al. 2020).

To improve run time, we simulated sister branches in parallel and recorded the final genealogies as tree sequences (Kelleher et al. 2016). Further, neutral mutations were not simulated with SLiM and were added after the fact with msprime. The resulting tree sequences were later joined and recapitated (i.e. we simulated genetic variation in the ancestor of all great apes using the coalescent) using msprime, tskit, and pyslim (Kelleher et al. 2016, 2018; Rodrigues and Ralph 2021). Despite our efforts to improve run time, our simulations of the entire history of the great apes were still incredibly costly (taking over a month to complete in many instances).

In our neutral simulations, we assumed that neutral mutations occurred at a rate of $2 \times 10^{-8}$ new mutations per generation per site (Scally and Durbin 2012), uniformly across the entire chromosome. To understand the effects of natural selection on landscapes, we simulated beneficial and deleterious mutations only within exons, assuming that the locations of exons were shared across all great apes (Kronenberg et al. 2018) and using exon annotations from the human reference genome NCBI36/hg18. We varied the proportions of neutral, beneficial, and deleterious mutations within exons, but the distribution of fitness effects (DFE) for both deleterious and beneficial mutations were shared across all apes. The DFE for deleterious mutations was gamma-distributed with a fixed shape $\alpha$ and scale as estimated in Castellano et al. (2019), and the DFE for beneficial mutations followed an exponential distribution (Orr 2003).

By default, we added neutral mutations to the simulated genealogies with msprime so that the total mutation rate (of neutral plus nonneutral mutations, if any) was constant along the genome. In addition, to simulate local variation in mutation rates along the chromosome, we selected 3 simulated genealogies (the fully neutral simulation, 1 with deleterious mutations and 1 with both beneficial and deleterious mutations) to add neutral mutations in a way that resulted in varying levels of neutral mutation rate variation along the chromosome. To do this, we built mutation rate maps by sampling mutation rates for each

**Table 1.** Range of parameters explored in the simulations.

| Regime | Neutral | Deleterious only | Beneficial only | Both |
|---|---|---|---|---|
| Proportion of deleterious mutations | 0% | 10–70% | 0% | 10–70% |
| Proportion of beneficial mutations | 0% | 0% | 0.005–0.5% | 0.005–0.5% |
| Deleterious DFE | – | Gamma distributed with $\bar{s}=$ $\{-0.015, -0.03\}$ and $\alpha = 0.16$ | – | Gamma distributed with $\bar{s} = \{-0.015, -0.03\}$ and $\alpha = 0.16$ |
| Beneficial DFE | – | – | Exponentially distributed with $\bar{s} = \{0.01, 0.005\}$ | Exponentially distributed with $\bar{s} = \{0.01, 0.005\}$ |

Nonneutral mutations were only allowed within exons. Gamma distribution was parameterized with shape $\alpha$ and mean $\bar{s} = \alpha/\beta$, where $\beta$ is the rate parameter. DFE, distribution of fitness effects.

1 Mb window independently from a normal distribution with mean $2 \times 10^{-8}$ and standard deviation chosen from $\sigma/2 \times 10^{-8} =$ $\{0.005, 0.007, 0.011, 0.016, 0.023, 0.033, 0.048, 0.070.0.103, 0.150\}$. In simulations with nonneutral mutations, we subtracted the nonneutral mutation rate from the respective window mutation rate for the intervals that intersected with exons. In total, we explored 56 different parameter combinations with all the different simulations (see Table 1 and Supplementary Table 1 for the parameter space). The code used to produce the simulations can be found at https://github.com/kr-colab/greatapes_sims.

## Visualizing correlated landscapes of diversity and divergence

To compare landscapes of diversity and divergence along chromosomes, we computed the Spearman correlation between the landscapes across windows within a chromosome. Because of computational constraints, we focus on chromosome 12. Chromosome 12 is one of the smallest chromosomes in the great apes, there are no major inversions, and it has good variation in exon density and recombination rate. The choice was made blindly before looking at the data, but we found it behaves similarly to other chromosomes (see Supplementary Figs. 10–31).

We expected landscapes of 2 closely related species to be more correlated than the landscapes of 2 distantly related species. Thus, the correlation between any 2 landscapes of diversity and divergence is expected to depend on distances between them in the phylogenetic tree. We decided to plot our correlations against distance (in generations) between the most recent common ancestor (MRCA) of each landscape. These distances were computed using the demographic model estimated in Prado-Martinez et al. (2013). In comparing 2 landscapes of diversity, this amounts to the total distance between the 2 tips in the species tree. For instance, the phylogenetic distance $dT$ between diversity in humans and diversity in bonobos is the sum of the lengths of the human, pan and bonobo branches in the species tree used for simulation (shown in Fig. 1). In comparing a landscape of diversity to a landscape of divergence, this amounts to the distance between the species of the landscape of diversity and the MRCA of the 2 species involved in the divergence. For example, $dT$ for the landscapes of diversity in humans and divergence between Sumatran orangutans and eastern gorillas would be the distance between the humans tip and the great apes internal node. $dT$ for the landscapes of divergence between the orangutans and divergence between the gorillas would be the distance between the orangutan and gorilla internal nodes. Some divergences may share branches in the tree, but these are excluded from our main figures; see the

Subsection Correlation between divergences that share branches in the Supplementary Material and Fig. 2.

## Results

First, we will provide a qualitative view of the landscapes of diversity and divergence in the great apes. Then, we explore the correlations between landscapes in the real data and how they vary depending on phylogenetic distance. To understand the processes that can drive these correlations, we use forward-in-time simulations of the great apes history under different models (e.g. with and without natural selection). Lastly, we describe how genomic features are related to patterns of diversity and divergence in the real great apes data, and we speculate which processes can explain what we see in the data and simulations.

### Landscapes of within-species diversity and between-species divergence

There is considerable variation in levels of genetic diversity across the great apes (Fig. 2). Species may differ in overall levels of diversity due to population size history: species with greater historical population sizes (e.g. central chimps and western gorillas) harbor the most amount of genetic variation (Prado-Martinez et al. 2013). Levels of diversity vary along the chromosome, but do not appear to be strongly structured. Instead, diversity seems to haphazardly fluctuate up and down along the chromosome, and this variation might be attributed to neutral genealogical and mutational processes alone. A notable feature is the large dip in diversity around the 50 Mb mark, which is so extensive that it almost erases the differences between-species. This dip coincides with 3 of the windows with the highest exon density, possibly pointing to the role of selection in shaping genetic variation in those windows.

Levels of between-species genetic divergence also vary along the genome, by an even greater amount in absolute terms. Interestingly, diversity ($\pi$) varies (along the chromosome) by about 0.2%, whereas divergence ($d_{XY}$) varies by more than 0.5%. Because $d_{XY} = \pi^{anc} + rT$ (where $\pi^{anc}$ is diversity in the ancestor, $r$ is the substitution rate, and $T$ is the split time between the 2 species), this excess in variance may be due to the substitution process. Landscapes of divergence which share their most common recent ancestor (e.g. human–Bornean orangutan and bonobo–Bornean orangutan divergences—both colored in red in Fig. 2a) overlap almost perfectly with each other. Curiously, divergence seems to accumulate faster in the ends of the chromosome, leading to a "smiley" pattern in the landscape of divergence—which is not apparent in the landscape of diversity. That is, with deeper split times, divergence in the ends

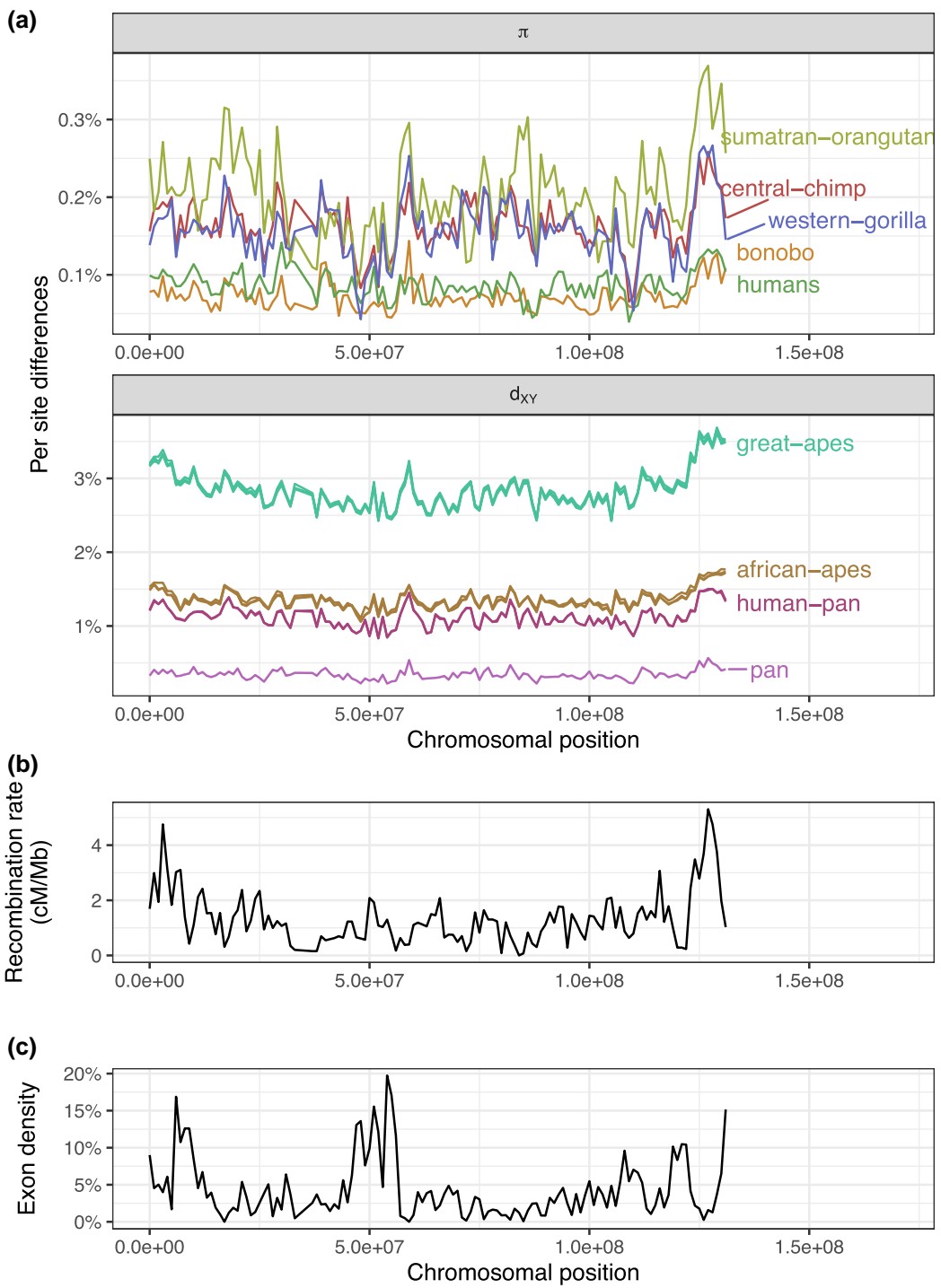

**Fig. 2.** a) Landscapes of nucleotide diversity ($\pi$) and divergence ($d_{XY}$) in 1 Mb windows along chromosome 12. Lines are colored by species on the top plot and by the MRCA on the bottom. Genomic windows with less than 40% of callable sites were masked. Only a subset of the species are displayed for clarity. b) Recombination rate estimates from humans (deCODE map; Kong *et al.* 2002). c) Exon density along chromosome 12, computed as the percentage of callable nucleotides in a window that fall within an exon.

of the chromosome seem to increase faster than in other regions of the genome (see how the divergences whose MRCA is the great apes look more like a convex parabola than a horizontal line in Fig. 2a; see also Supplementary Fig. 1).

In comparing landscapes across species side by side, a remarkable pattern emerges: levels of genetic diversity and divergence

along chromosomes have similar peaks and troughs. To get a sense of how strong this observation is, we can compare it to one of the most well-studied properties of genomic variation: the correlation between exon density and genetic diversity. We found that the correlation between human diversity and exon density is −0.2 (at the 1 Mb scale), but the correlation

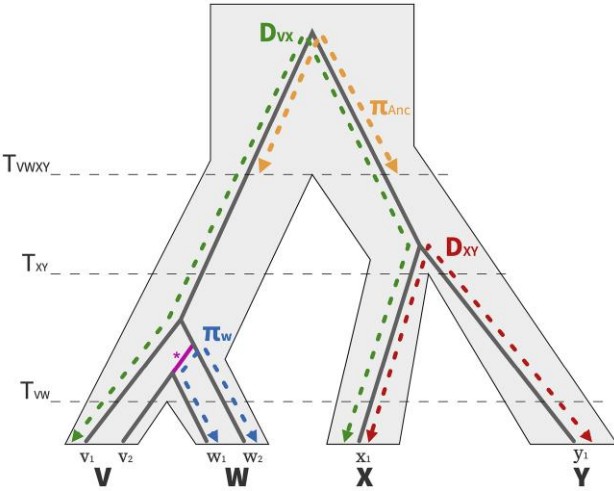

**Fig. 3.** Visualizing the relationships between nucleotide diversity and divergence statistics between closely related taxa. A population and gene tree for 4 populations (*V, W, X, Y*) are depicted with the light gray polygon and gray solid line, respectively. The branch that is shared between $\pi_V$ and $\pi_W$ due to ILS is highlighted in pink.

between levels of diversity in humans and western gorillas is 0.48. Below, we dissect this observation of strong correlation between landscapes across the great apes and discuss the processes that may cause it.

## Remarkable correlations between landscapes of diversity and divergence

The landscapes of diversity and divergence are highly correlated across the great apes. To interpret this signal, we first need to understand what processes can cause such correlations, and so first we describe the toy example depicted in Fig. 3. Both genetic diversity ($\pi$) and divergence ($d_{XY}$) are estimates of the mean time to the most recent common ancestor (multiplied by twice the effective mutation rate). Populations V and W split recently, and so ancestral variation contributes significantly to within-species diversity (i.e. the coalescences for samples within species happen before the species split). As a result, samples from 1 population may coalesce first with a sample from another species (e.g. samples $v_2$ and $w_1$), a pattern called ILS (see the branch marked with * in the gene tree). This sharing of ancestral variation causes $\pi_V$ and $\pi_W$ to be correlated with each other. The probability 2 samples from V coalesce before the split with W is $1 - e^{\frac{-T}{2N_e}}$, where $T$ is the split time and $N_e$ is the effective population size. Therefore, split time ($T$) should be a good predictor of the correlation between 2 landscapes of diversity (and/or divergence). Thus, we decided to visualize correlations between landscapes of diversity and divergence by computing the phylogenetic distance $dT$, which is simply the distance in generation time between 2 statistics. For example, we define $dT(\pi_W, d_{XY}) = 2T_{VWXY} - T_{XY}$. Divergences may share branches by definition (irrespective of split times), as you can see with $d_{VX}$ and $d_{XY}$ (see Subsection Visualizing correlated landscapes of diversity and divergence for more details). In such cases, our chosen metric $dT$ would not be a good proxy for expected correlations, so we omit such cases from our main figures. See Subsection Visualizing correlated landscapes of diversity and divergence and Supplementary Fig. 2 for more on the correlations between landscapes that share branches.

Figure 4 shows the pairwise correlations between great apes landscapes of diversity and divergence against phylogenetic distance ($dT$, which is computed from the split times of the model shown in Fig. 1). We see ancestral variation seems to play a role in structuring correlations between landscapes: pairs of species that recently split have their landscapes of diversity highly correlated. Surprisingly, correlations still plateau at around 0.5. We expect ancestral variation to play a minor role when comparing orangutans and chimps, which separated around $60 \times N_e$ generations ago, but their landscapes are still highly correlated. Population size history seems to affect the correlation between landscapes since the weakest correlations involve the landscape of diversity of one of the species with small historical population sizes (i.e. bonobos, eastern gorillas, and western chimps).

Correlations between landscapes of divergence and diversity and between landscapes of divergence are also quite high, often surpassing 0.5, and they also decay with phylogenetic distance ($dT$) (see middle and right most plots in Fig. 4). In theory, these landscapes can also be correlated due to ancestral variation. To see how ancestral variation can create correlations even between landscapes with no overlap in the tree, consider Fig. 3: divergence between X and Y and divergence between V and W can each contain contributions from ancestral diversity if lineages have not coalesced in both branches leading from the ancestor. If a particular portion of the genome happens to have higher diversity in the ancestor, it will also have higher divergence. Since this correlation is produced by sharing of ancestral variation, it is expected to have a very small effect except when branches are short. As discussed in Subsection Visualizing correlated landscapes of diversity and divergence, 2 divergences can also be correlated by definition (because they share branches in the tree). For example, when comparing human–Bornean orangutan and gorilla–Bornean orangutan divergence we expect some correlation because these divergences share the large African apes and orangutan branches in the tree (Fig. 1). In Fig. 4, we excluded these comparisons where branches are shared. Such comparisons can be seen in Supplementary Fig. 2. We found that even these comparisons that share branches have an excess of correlation compared to a theoretical expectation (derived from a simplified neutral model), that is the correlations are above the $y = x$ line in Supplementary Fig. 2 even for distantly related species.

There are many processes that could maintain landscapes correlated. Above, we discussed how we expect ancestral variation to explain these correlations. The alternative would be to have a process that structures variation along chromosomes which is shared across species. Using forward-in-time simulations, we set out to (1) confirm that ancestral variation alone is not causing landscapes to remain correlated and (2) test which process or processes that when shared among a group of species could maintain correlations in similar ways to what we observed in the great apes' data.

## Neutral demographic processes

To assess the extent to which ancestral variation alone could explain our observations, we performed a forward-in-time simulation of the great apes' evolutionary history. As expected, the resulting landscapes of diversity and divergence are not well correlated (Fig. 5). Ancestral variation seems to maintain correlations between some landscapes; for instance, the landscapes of diversity in central and eastern chimps have a 0.61 correlation, the highest across all pairs of comparisons (Fig. 5a, point a). Nevertheless, correlations between landscapes of diversity and divergence decay quickly with phylogenetic distance to 0. Some distant comparisons are moderately correlated (e.g. the landscape of

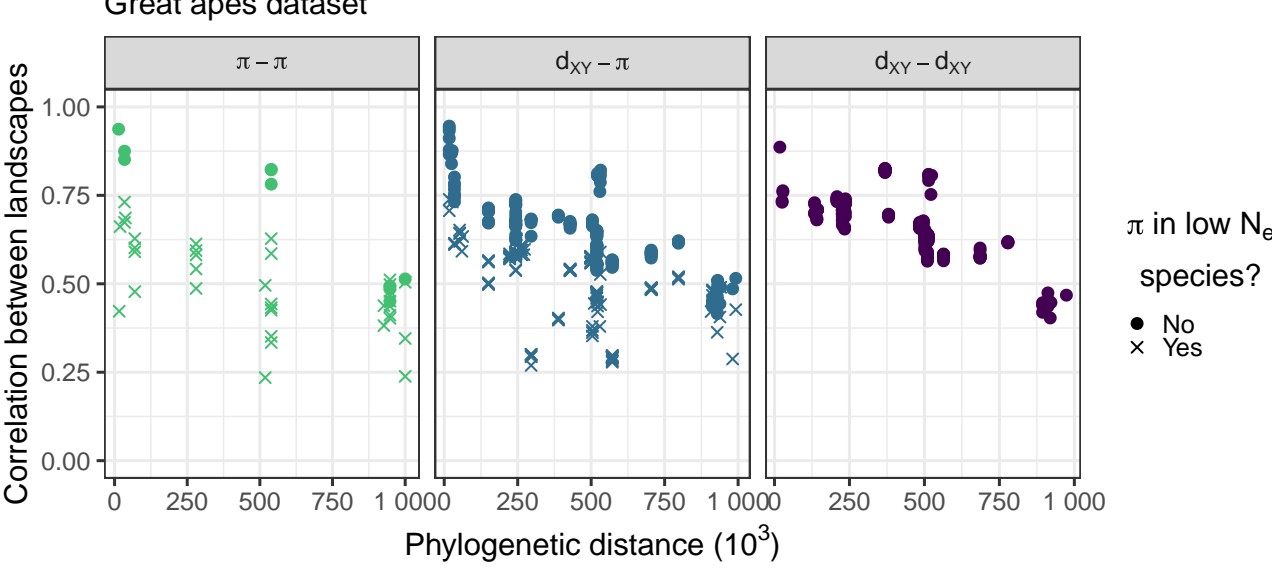

**Fig. 4.** Correlations between landscapes of diversity and divergence across the great apes. Each point on the plots correspond to the (Spearman) correlation between 2 landscapes of diversity/divergence, computed on 1 Mb windows across the entire chromosome 12. Correlations were split by type of landscapes compared ($\pi$–$\pi$, $\pi$–$d_{XY}$, $d_{XY}$–$d_{XY}$). $dT$ is the phylogenetic distance (in number of generations) between the most common recent ancestor of the 2 landscapes compared (e.g. the $dT$ for correlation between landscapes of diversity in humans and divergence between eastern gorillas and orangutans is distance between the humans and the great apes nodes in the phylogenetic tree, Fig. 1). Note that species with low $N_e$—for which the estimated species $N_e$ was less than $8 \times 10^3$: bonobos, eastern gorillas, and western chimps—have a different point shape. Only comparisons for which the definition of the statistics do not overlap are shown, as explained in Subsection Visualizing correlated landscapes of diversity and divergence.

**Fig. 5.** Landscapes are not well correlated in a neutral simulation. a) Correlations between landscapes of diversity and divergence in a neutral simulation. See Fig. 4 for more details. b) Nucleotide diversity and divergence along the simulated neutral chromosome. See Fig. 2a for details.

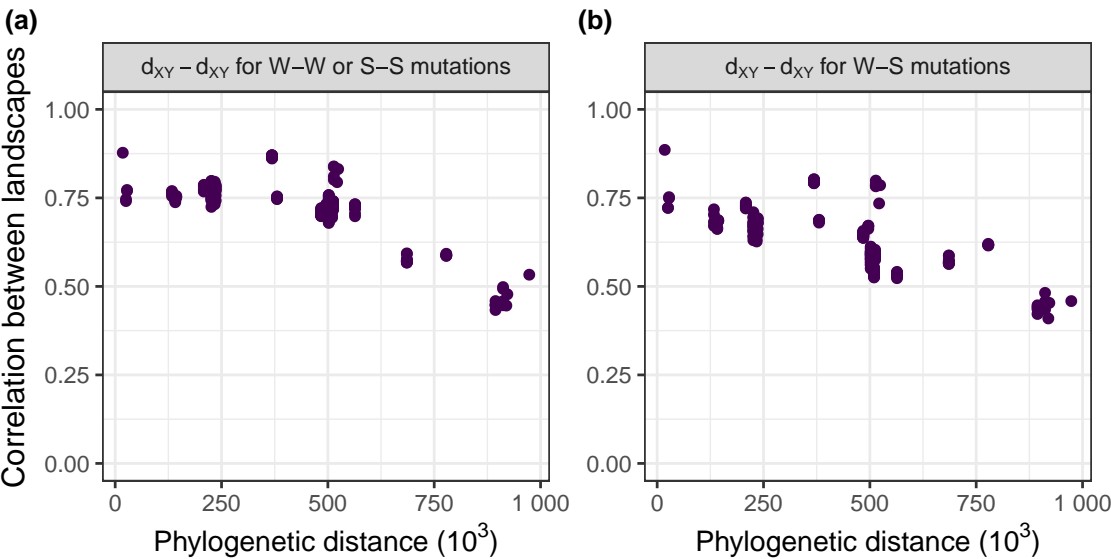

**Fig. 6.** Correlations between landscapes of divergence partitioned by site type (W–W/S–S and W–S). W–W sites are sites in which the state did not change between-species (and remained weak which corresponds to A or T). Similar logic applies to S–S sites (S or strong states are G or C). W–S sites are sites in which a new mutation appeared either going from weak to strong or from strong to weak. Note these definitions do not rely on identifying the exact ancestral state, we simply compare the current states in the 4 species involved (2 species per $d_{XY}$ landscape). For example, if by looking at the 4 species, we see the following states A, T, A, T the site would be classified as W–W. If we saw G, A, A, A, the site would be classified as W–S. Other details are the same as in the rightmost panel in Fig. 4.

diversity in Bornean orangutans and divergence between central and western chimps have a correlation coefficient of 0.23, see Fig. 5a, point b), but that seems to be driven by the outlier window around 80 Mb. This outlier window has a recombination rate close to 0 (Fig. 2c), so the average nucleotide diversity over the window has a higher variance because of coalescent noise (see the extreme peaks and valleys in Fig. 5). Recombination rate variation can create some moderate correlations, but when we look at multiple species at once it becomes clear that the mean correlation goes to 0.

### GC-biased gene conversion

A prominent feature of the landscapes of divergence in the great apes is the faster accumulation of divergence in the ends of the chromosomes (Fig. 2). This feature was not present in any of our simulations, so we sought to understand its possible causes. Double-stranded breaks are more common at the ends of chromosomes (Kong et al. 2002, 2010), and these can be repaired either by crossover or gene conversion events. GC-biased gene conversion (gBGC), the process whereby weak alleles (A and T) are replaced by strong alleles (G and C) in the repair of double-stranded breaks in heterozygotes, mimics positive selection—in that it increases the probability of fixation of G and C alleles (e.g. Galtier et al. 2009). We suspected gBGC could have caused the increased rate of accumulation divergence in the ends of chromosomes, as has been observed previously (Katzman et al. 2010), and contributes to the maintenance of correlations between landscapes over long time scales.

To tease apart the effects of gBGC on correlated landscapes, we partitioned divergence by mutation type (weak to weak, strong to strong, and weak to strong). If correlations are being driven by gBGC, then we would expect the correlation between landscapes of divergence to be stronger for weak to strong mutations. We found that the overall correlations are very similar across mutation types, suggesting gBGC does not play a strong role in structuring the correlations between landscapes (Fig. 6).

### Positive and negative natural selection

Another process whose intensity is likely correlated across all branches in the great apes tree is natural selection. If targets of selection and recombination maps are shared across species, then we would expect both the direct and indirect effects of selection to be shared across branches. It can be difficult to model natural selection in a realistic manner because we do not know precisely which locations of the genome are subject to stronger selection. Nevertheless, exons are expected to have higher density of functional mutations than other places in the genome. Thus, we ran simulations in which beneficial and deleterious mutations can happen only within exons. Using human annotations, we simulated the great apes' history assuming a common recombination map and exon locations. See the landscapes resulting from these simulations in Supplementary Fig. 3.

We found that negative selection can slightly increase correlations between landscapes (Fig. 7a–c). If 30% of all mutations within exons were strongly deleterious (mean selection coefficient $\bar{s} = -0.03$), landscapes would be weakly correlated (Fig. 7b). The correlations between landscapes rarely surpass 0.5, even with 70% of all mutations within exons being strongly deleterious (Fig. 7c).

Positive selection, on the other hand, can quickly increase correlations between landscapes. A beneficial mutation rate within exons of $\bar{\mu}_p = 1 \times 10^{-12}$ produced moderate correlations between landscapes (Fig. 7d). With too much positive selection, correlations can break down because of the contrasting effects of positive selection on diversity and divergence. That is, while positive selection increases fixation rates and hence divergence between-species, its linked effects decrease diversity within the species. This can create negative correlations between landscapes, as can be seen in Fig. 7f. Note that some correlations between landscapes of diversity and divergence remain high when the divergence is computed between closely related species (e.g. central and eastern chimps). Divergence is $d_{XY} = \pi^{anc} + 2rT$, where $\pi_{anc}$ is diversity in the ancestor, $r$ is the substitution rate, and $T$ is the

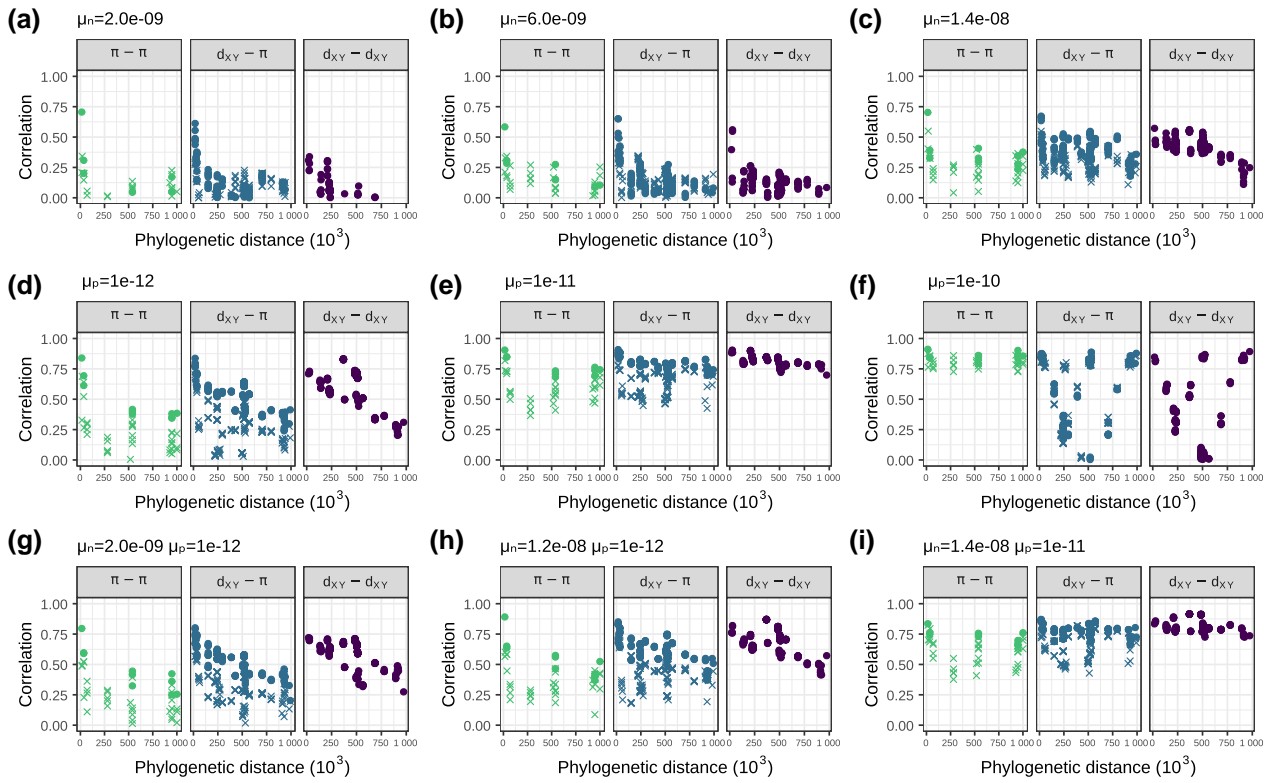

**Fig. 7.** Correlations between landscapes of diversity and divergence in simulations with natural selection. a–c) Simulations with negative selection. d–f) Simulations with positive selection. g–i) Simulations with both negative and positive selection. The selection parameters $\mu_n$ and $\mu_p$ are the rate of mutations in exons with negative and positive fitness effects, respectively. The mean fitness effect was $\bar{s} = -0.03$ for deleterious mutations and $\bar{s} = 0.01$ for beneficial mutations (see Subsection Simulations for more details). See how panel h looks the most like the data (Fig. 4).

time since species split. Thus, for the divergences in which the 2 species split recently are dominated by genetic diversity in the ancestor, correlations between $\pi$– $d_{XY}$ remain high because $d_{XY} \simeq \pi^{\mathrm{anc}}$.

Positive and negative selection can work synergistically to produce correlated landscapes that look like the real data. For example, comparing Fig. 7d, g, h which differs in rate of negatively selected mutations $\mu_n$, it is possible to see that the correlations between landscapes start to resemble the real data with more deleterious mutations. Figure 7h seems to resemble the data fairly well, with $\pi$– $d_{XY}$ and $d_{XY}$– $d_{XY}$ correlations plateauing around 0.5. The $\pi$–$\pi$ correlations are a bit lower than the real data, however. Recent demographic events can affect genetic diversity and although our simulations are heavily parameterized with respect to the effects of selection, we are not capturing all the variation caused by more realistic demographic models. Figure 7d and h look very similar to each other. These have the same amount of positive selection, but the first did not have any negative selection. The major difference between them is that with negative selection there is a more clear separation between the correlations involving low $N_e$ species, similar to what is seen in the data.

## Mutation rate variation

Since mutation rate can vary along chromosomes, if this mutation rate map were shared across species, it would maintain correlations between landscapes over longer periods of time. To assess this, we used 3 of our previous simulated genealogies of the great apes and replaced all neutral mutations assuming a common neutral mutation rate map across the phylogeny: for each

window, we drew a mutation rate from a normal distribution with mean $2 \times 10^{-8}$ (the same as all other simulations) and standard deviation $\mu_{SD}$. We found that, under neutrality, a mutation rate map with $\mu_{SD}$ close to $7\% \times 2 \times 10^{-8}$ would be needed to get correlations similar to the data (Fig. 8a–c). Although mean correlations look similar to the data, we see that correlations tend to increase slightly with time in the simulations with mutation rate variation. This is expected because windows with higher mutation rate accumulate divergence faster, creating a correlation with mutation rate that gets stronger with time. In the great apes' data, however, we see a slow but steady decrease in correlations with time.

When we added variation in the neutral mutation rate to simulations with selection, we found that a mutation rate map with a standard deviation of rates of slightly less than $\mu_{SD} = 7\%$ could plausibly create the correlations observed in the real data (Fig. 8d–i). The neutral and deleterious simulations with mutation rate variation fail to recover 1 aspect of the real data: the lower correlations between landscapes that include at least 1 low $N_e$ species (seen in the $\pi$–$\pi$ and $\pi$– $d_{XY}$ comparisons of Fig. 4). This feature, however, is seen in the simulations with both beneficial and deleterious mutations (Fig. 8g–i).

## Visualizing similarity between simulations and data

To see how a particular simulation resembles the real data, we can use Figs. 4 and 7 to compare how the patterns of all 1,260 pairwise correlations between landscapes match the real data. However, it is difficult to assess the fit of the simulated scenarios

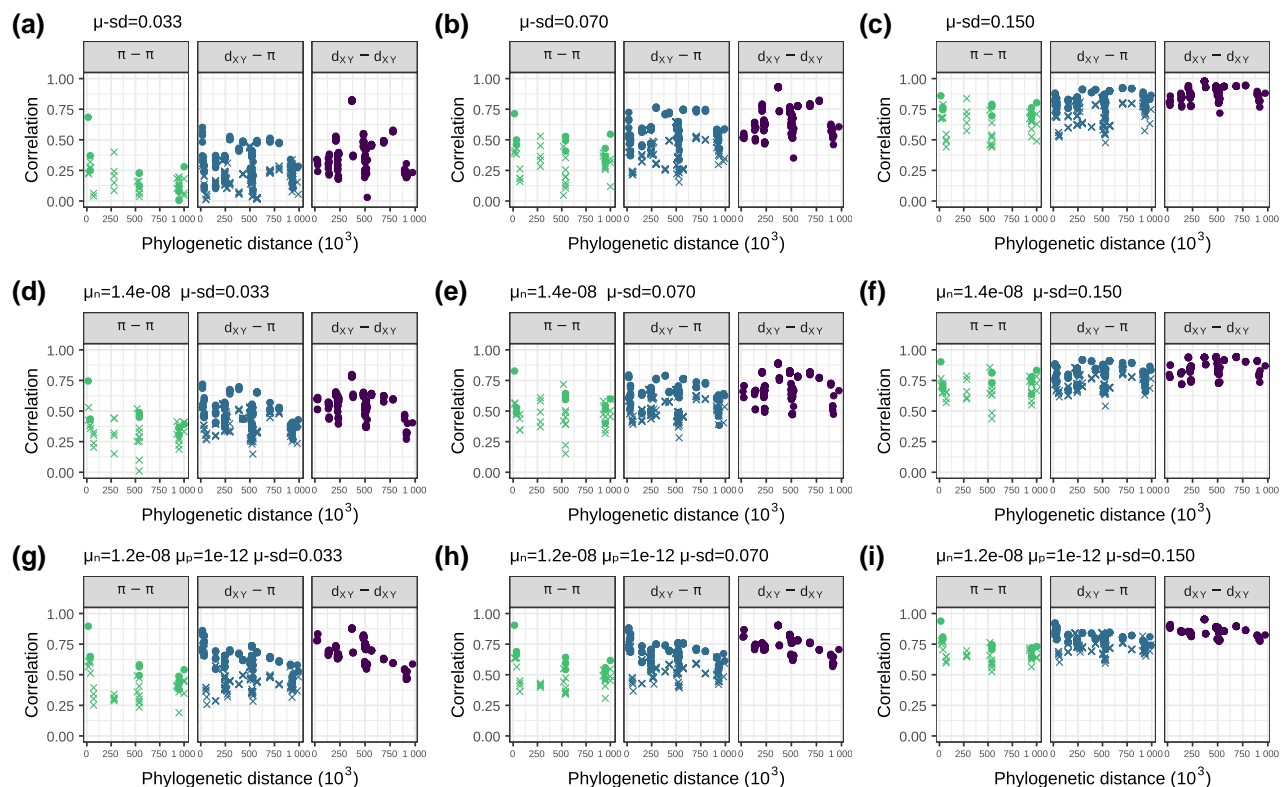

**Fig. 8.** Correlations between landscapes of diversity and divergence across the great apes for simulations with variation in mutation rate along the chromosome. Panels a–i show different simulations in which we varied the standard deviation in neutral mutation rate between 1 Mb windows, in each setting the standard deviation to the mean mutation rate ($2 \times 10^{-8}$) multiplied by $\mu_{SD}$. First row a–c) use a neutral simulation, second row d–f) a simulation with negative selection, and third row g–i) a simulation with both positive and negative selection. The selection parameters $\mu_n$ and $\mu_p$ are the rate of mutations in exons with negative and positive fitness effects, respectively. The mean fitness effect was $\bar{s} = -0.03$ for deleterious mutations and $\bar{s} = 0.01$ for beneficial mutations (see Subsection Simulations for more details). See how without selection a–c), the simulation with $\mu_{SD} = 7\%$ (panel b) looks close to the data (Fig. 4). With selection, the simulation with both positive and negative selection and $\mu_{SD} = 3.3\%$ looks even more similar to the data (correlations between divergences decay over time, and there is a more pronounced differentiation between low and high $N_e$ comparisons).

to real data from such a comparison. Instead, we use principal component analysis (PCA) and create a low-dimensional visualization, shown in Fig. 9, in which each point is a simulation or the real data (shown in yellow). We created this PCA from the 57 × 1, 260 matrix in which rows are the simulations and the data, and columns are the pairwise Spearman correlations between landscapes. Unlike in the plots above, here we include the correlations between overlapping landscapes (as detailed in Subsection Visualizing correlated landscapes of diversity and divergence) (Fig. 9). In PC space, the data most closely resembles a subset of our simulations with both positive and negative selection (e.g. $\bar{\mu}_p = 1 \times 10^{-12}$ and $\bar{\mu}_n = 1.2 \times 10^{-8}$), including no or very little variation in mutation rates (less than 4%).

We also performed PCA on correlations computed at 2 different scales, 500 Kb and 5 Mb, in addition to the previously shown results for 1 Mb (Supplementary Figs. 5 and 6). At 500 Kb, the observed data are slightly more distant from simulations than at the higher scales, possibly because the recombination map used in simulations had a coarser resolution. Nevertheless, the observed data most closely match the simulations with both positive and negative selection in all scales.

## Correlations between genomic features and diversity and divergence

Next, we describe how 2 important genomic features (i.e. exon density and recombination rate) are related to diversity and

divergence in the real great apes dataset. The correlations between recombination rate and genetic diversity are positive in all great apes (Fig. 10a). The strongest correlation between genetic diversity and recombination rate is seen in humans, which is unsurprising given our recombination map was estimated for humans. Recent demographic events also seem to impact the strength of the correlation; for example, the correlation between recombination rate and diversity is higher in Nigerian chimps than in western chimps, which have a much lower recent effective population size. We found that diversity is negatively correlated with exon density across all species (Fig. 10d). Contrary to what we observed with recombination rate, the correlation between exon density and diversity was even stronger in most other apes than in humans. Species with smaller $N_e$ tend to show weaker correlation between diversity and exon density (see Nam *et al.* 2017 for related findings). A striking feature of the correlations of between-species divergence and genomic features, shown in Fig. 10, is that the correlations get stronger with the amount of phylogenetic time that goes into the comparison (i.e. the $T_{MRCA}$), in a way that is roughly linear with time.

To describe why this increase in correlation with time might occur, we turn to an analytic approach. Genetic divergence (*D*) in the ith window between 2 species that split *t* generations ago can be decomposed as

$$D_i(t) = \pi_i(t) + R_i t + \varepsilon_i,$$

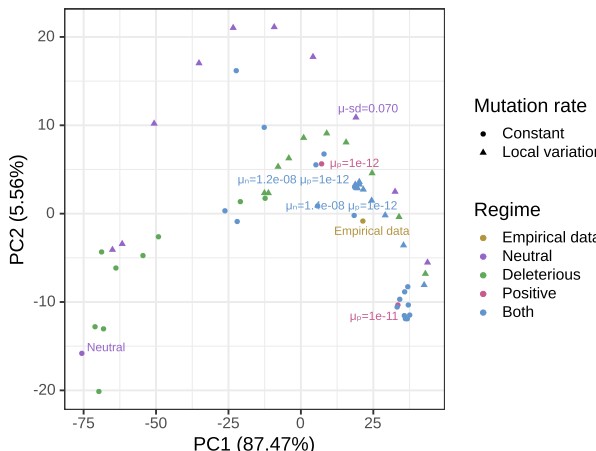

**Fig. 9.** PCA visualization of data and simulations. Colors differentiate the empirical data from simulations with different parameters: "Neutral" refers to the simulation without any selection, "Deleterious" refers to simulations with deleterious mutations, "Positive" refers to simulations with beneficial mutations, "Both" refers to simulations with both beneficial and deleterious mutations. The shape of the points differentiate simulations with constant mutation rate along the genome and variable local mutation rates. Local variation in mutation rates were added on top of 3 simulations: neutral, deleterious with $\mu_n = 1.4 \times 10^{-8}$, and both with $\mu_n = 1.2 \times 10^{-8}$ and $\mu_p = 1 \times 10^{-12}$. The PCA performed on a matrix containing all pairwise correlations between landscapes across the great apes (i.e. all $\pi$–$\pi$, $\pi$–$d_{XY}$, and $d_{XY}$–$d_{XY}$ comparisons) for the great apes dataset and simulations (with selection and with mutation rate variation). We excluded simulations with $\mu_p \geq 1 \times 10^{-10}$ from the PCA analysis because PC2 was capturing negative correlations caused by strong positive selection—as seen in Fig. 7f.

where $\pi_i(t)$ is the genetic diversity in the ancestor at time $t$, $R_i$ is the substitution rate in the window and $\varepsilon_i$ is a contribution from genealogical and mutational noise (which has mean zero). This decomposition follows from the definition of genetic divergence as the number of mutations since the common ancestor, as depicted in Fig. 3 (see how $D_{VX} = \pi^{\mathrm{anc}} + 2RT_{VWXY}$).

The covariance between $D(t)$, the vector of divergences along windows, and a genomic feature $X$ is, using bilinearity of covariance,

$$\mathrm{Cov}(D(t), X) = \mathrm{Cov}(\pi(t), X) + t\mathrm{Cov}(R, X) + \mathrm{Cov}(\varepsilon, X). \qquad (1)$$

Happily, this equation predicts the linear change of the covariance with time that is seen in Fig. 10c and perhaps Fig. 10f. However, caution is needed because the correlation between diversity and the genomic feature ($\mathrm{Cov}(\pi(t), X)$) may be different in different ancestors, and indeed the inferred effective population size is greater in older ancestors in the great apes (Fig. 1).

Next consider covariances of diversity with recombination rate, Fig. 10c. Consulting the equation above, the fact that the covariance between divergence and recombination rate increases with time can be caused by 2 factors (taking $X$ to be the vector of mean recombination rates along the genome): (1) a positive covariance between substitution rates and recombination rates ($\mathrm{Cov}(R, X) > 0$) and/or (2) greater genetic diversity in longer ago ancestors ($N_e(t)$ larger for larger $t$). It is unlikely that the increase in $N_e$ in more ancient ancestors was sufficient to produce the dramatic increase in covariance seen in Fig. 10c, since it would require $\mathrm{Cov}(\pi(t), X)$ to be far larger in the ancestral species than is seen in any modern species. On the other hand, there are various plausible mechanisms that would affect $\mathrm{Cov}(R, X)$. One factor that certainly contributes is the "smile": we found that divergence

increases faster near the ends of the chromosomes where recombination rate is greater, probably in part because of GC-biased gene conversion. Interestingly, positive and negative selection are predicted to have opposite effects here: greater recombination rate increases the efficacy of both through reduced interference among selected alleles, so positive selection would increase substitution rate and hence increase $\mathrm{Cov}(R, X)$, while negative selection would decrease $\mathrm{Cov}(R, X)$. When considering only the middle half of the chromosome (i.e. excluding the effect of gBGC) (Supplementary Fig. 7), the covariances between divergence and recombination rate flip to negative, and they continue to decrease over time. Thus, it seems that negative selection is the most important driver of divergence in the middle, whereas gBGC strongly affects the tails of the chromosome.

The covariance of diversity and exon density has a less clear pattern (Fig. 10f), although it generally gets more strongly negative with time. This decrease could be a result of a negative covariance between substitution rates and exon density and/or an increase in the population sizes of the ancestors (if $\mathrm{Cov}(v, X) < 0$, as expected since $v$ is relative diversity and $X$ is now exon density). As before, positive selection in exons would be expected to produce a positive covariance between exon density and substitution rate, while negative selection would produce a negative covariance. It is hard to determine a priori which is likely to be stronger, because although negative selection is thought to be much more ubiquitous, a small amount of positive selection can have a strong effect on substitution rates. The fact that covariance generally goes down with time suggests that negative selection (i.e. constraint) is more strongly affecting substitution rates.

It is at first surprising that the correlations between exon density and divergence go up with time, but the covariances go down with time (Fig. 10e, f). However, correlation is defined as $\mathrm{Cor}(D_t, X) = \mathrm{Cov}(D_t, X)/\mathrm{SD}(D_t)\mathrm{SD}(X)$. Thus, if the variance in divergences increases over time the correlations will decrease over time. Indeed, we see this happening as gBGC increases divergences on the ends of the chromosome faster than in the middle, leading to an increase in variance of divergence along the genome. This also explains why correlations of landscapes of very recent times are very noisy, but covariances are not. Indeed, the patterns are clearer when we exclude the tails of the chromosome (Supplementary Fig. 7): there is only a modest increase in the correlation between exon density and divergence over time and the covariances go down with time more linearly.

## Discussion

A central goal of population genetics is to understand the balance of evolutionary forces at work in shaping the origin and maintenance of variation within and between-species (Lewontin 1974). While the field has been historically data-limited, with the current flood of genome sequencing data, we are poised to make progress on such old questions. Over the past decades, an important lever in understanding the relative impact of genetic drift vs selection in shaping genomic patterns of variation has been to examine the relationship between *levels* of diversity and genomic features, such as recombination rate and exon density. The overarching observation has been that regions of reduced crossing over generally harbor less variation than regions of increased crossing over in many but not all species (e.g. Begun and Aquadro 1992; Corbett-Detig et al. 2015). This observation is consistent with a role for linked selection shaping patterns of variation in recombining genomes, but the relative contributions of deleterious and beneficial mutations is still largely unknown. Indeed, it seems likely that some complex

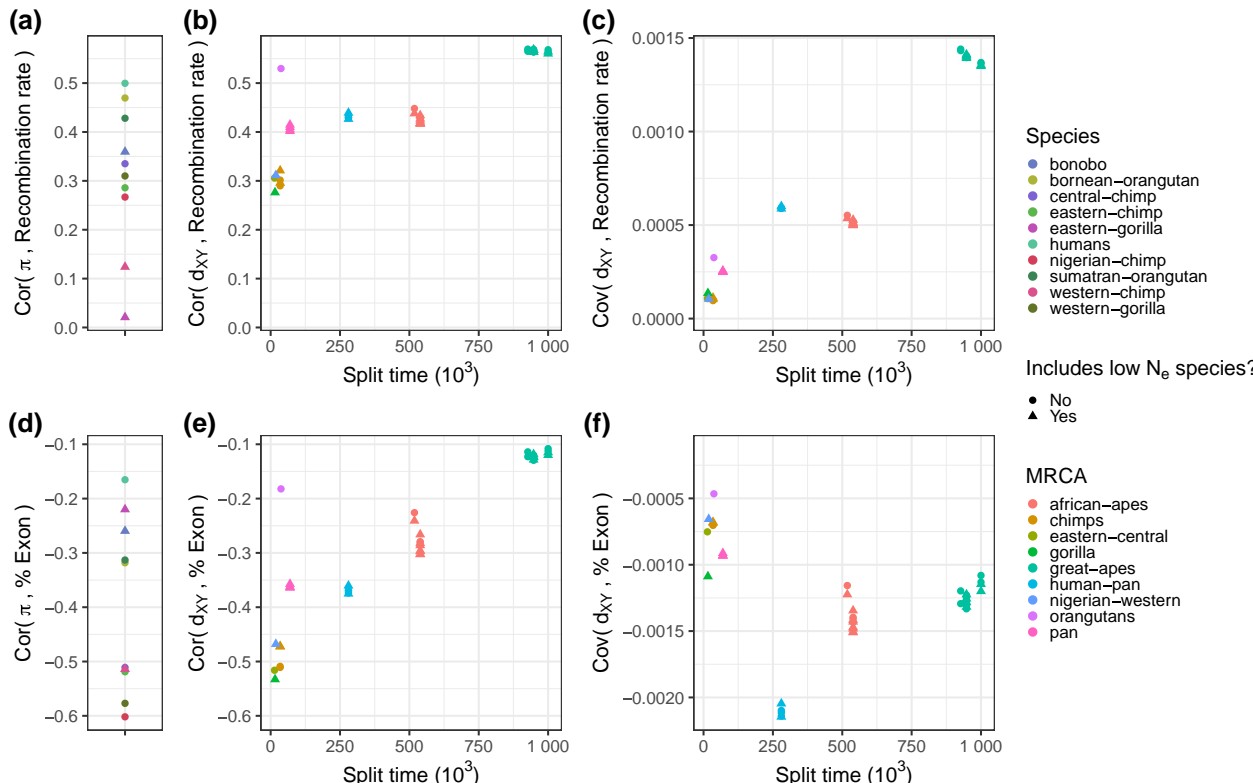

**Fig. 10.** Correlations and covariances between landscapes of diversity and divergence and annotation features in the real great apes data. Exon density and recombination rates were obtained as detailed in Fig. 2. Split time is the time distance between the 2 species involved in the divergence. Points are colored by the species of within-species diversity ($\pi$) in plots a and d. In plots b, c, e, f, the points are colored by the most common recent ancestor of the species for which between-species divergence was computed. Species with low $N_e$—for which the estimated species $N_e$ was less than $8 \times 10^3$: bonobos, eastern gorillas, and western chimps—have a different point shape.

mixture of both processes shapes variation in natural populations (Kern and Hahn 2018).

In this paper, we moved beyond genetic diversity within a single species to look at how divergence between closely related species changes with time and how this correlates with genomic features. Previous studies (e.g. Stankowski *et al.* 2019) looked at similar patterns (in monkeyflowers) and found strong correlations between landscapes of diversity and divergence between related species, despite deep split times. Landscapes of closely related species can remain correlated for 2 main reasons: (1) shared ancestral variation or (2) shared heterogeneous process. If 2 species recently split, their landscapes of diversity are expected to be correlated due to shared ancestral variation. If the process that structures genetic diversity along chromosomes is heterogeneous and somewhat shared between-species, then their landscapes are expected to remain correlated over longer periods of time. For example, if the effects of selection are concentrated in the same genomic regions in 2 species, then their landscapes of diversity will be correlated. By incorporating information from multiple species at once, we are able to pool information across species and thus increase our power to disentangle the role of different evolutionary forces. Patterns across multiple species are more likely to be robust to the idiosyncrasies of any 1 species, such as demographic history. For instance within-species metrics can be confounded by demography: demographic events can create spurious troughs of diversity (Simonsen *et al.* 1995) or exacerbate the effects of background selection on diversity (Torres *et al.* 2018). However, correlations between landscapes can only be produced due to shared ancestral variation or a shared heterogeneous process.

In the great apes, we found that landscapes of within-species diversity and between-species divergence are highly correlated across the phylogeny. Those correlations are often stronger than those that have been historically used as evidence for the effects of selection on genetic variation. For example, the correlation between genetic diversity in humans and exon density is −0.2, yet the correlation between diversity in humans and diversity in western gorillas is 0.48. This stronger correlation may not be entirely due to shared landscape of selection—it may also be a result of shared ancestral variation (and ILS), mutation rate variation, and/or GC-biased gene conversion. To understand how much of the correlation between landscapes can be attributed to ancestral variation, we performed extensive simulations of the great apes' evolutionary history, and found that ancestral variation explains very little of the correlations we observed. Thus, a shared heterogeneous process seems to be needed to explain the data.

Two neutral processes can be heterogeneous along the genome and shared across species: GC-biased gene conversion and mutation. GC-biased gene conversion (gBGC) is thought to be an important factor in shaping levels of variation in humans (Chen *et al.* 2007; Glémin *et al.* 2015; Pouyet *et al.* 2018), and it has similar effects to those of natural selection. However, if gBGC were a major driver of correlations we would expect to see a difference in overall levels of correlation between different classes of substitution, and we do not (Supplementary Figs. 4 and 6). As such gBGC seems to be a minor contributor to the correlations we observe, although it does seem to be leading to increased substitution rates near the telomeres (where divergences are increasing roughly 5% faster; see Fig. 2 and Supplementary Fig. 1). In birds, an excess of

divergence near telomeres has been attributed to meiotic drives (Ellegren *et al.* 2012).

When the history of the great apes is simulated with a shared heterogeneous mutation map, correlations between landscapes do emerge. These were as strong as seen in the data when the rates were drawn from a normal distribution with a standard deviation of the mutation rate of at least a 7% of the mean mutation rate. However, our mutation map was perfectly shared among was species in our simulations, so it is possible that a mutation map which changes over time might move closely to match the data. Smith *et al.* (2018) estimated the standard deviation of de novo mutation rate in humans at the 1 Mb scale to be around 25% of the mean mutation rate. However, the lack of congruency in de novo mutations identified in different datasets raises questions about the role of ascertainment biases that need to be addressed in future studies (Castellano *et al.* 2020). Our simulations showed a facet of shared mutational heterogeneity along the genome that we do not observe in real data: with variable mutation rate correlations increase over time, whereas in the real data they decrease. It is unknown how conserved mutation rate heterogeneity is across the great apes, so it remains to be seen how an evolving heterogeneous mutation rate map affects landscapes of diversity and divergence. A major driver of mutation rate variation stems from CpG dinucleotides, which have much higher mutation rates than other sites (Nachman and Crowell 2000; Hodgkinson and Eyre-Walker 2011; Agarwal and Przeworski 2021). Nevertheless, when we partitioned the landscapes of divergence by mutation types, we did not see an excess of correlation between landscapes with mutations that can be affected by CpG-induced mutation rate variation (Supplementary Figs. 4 and 6).

Natural selection can also structure genetic variation heterogeneously along the genome. In simulations, both positive and negative selection are needed for the correlations between landscapes to resemble the data. By examining the correlations between landscapes (summarized in Fig. 9), we found that the best fitting simulation is the one with a beneficial mutation rate within exons of $1 \times 10^{-12}$ and deleterious rate within exons of $1.2 \times 10^{-8}$. Positive selection seems to be needed to explain 1 particular feature of the data: the separation between correlations involving a low $N_e$ species (i.e. correlations are lower if diversity is computed in a species with a low $N_e$—as seen in humans, bonobos, western chimps, and eastern gorillas; see Fig. 4). Bottlenecks can erase sweep signatures (Przeworski 2002; Jensen *et al.* 2005; Nielsen *et al.* 2005), but demography does not affect local variation in mutation rates, and it can exacerbate signatures of background selection (Torres *et al.* 2018). Thus, if sweeps are causing correlations between landscapes, we expect it to be more sensitive to the strong bottleneck in humans than the other processes. This conclusion largely agrees with previous studies which found that positive selection is necessary to explain reduction in genetic diversity surrounding genes in the great apes (Nam *et al.* 2017).

Another way we might characterize our simulations is through examination of substitution processes. In our best fitting simulation, we get a fixation rate of beneficial mutations of around $1 \times 10^{-9}$ per generation per exon base pair, what amounts to approximately 9% of the fixations within exons (along the human lineage), i.e. about 1 new fixation of a beneficial mutation every 250 generations. Fixation rate within exons is decreased to around 60% of the rate in our neutral simulation due to the constant removal of deleterious mutations within these regions. Indeed, previous studies (Boyko *et al.* 2008; Laval *et al.* 2021; Zhen *et al.* 2021) have estimated that between 10 and 16% of amino acid differences between humans and chimpanzees were caused by positive selection, which is strikingly similar

to our best fitting simulation. We would expect to see the fixation of around 16 beneficial mutations in the past 4,000 generations, which is close to the number of hard sweeps genome scans for selections have found in humans over this same time period (Schrider and Kern 2016, 2017). Our best fitting simulation with selection assumes that 60% of new mutations within exons are deleterious, similar to estimates from the site frequency spectrum (Boyko *et al.* 2008; Huber *et al.* 2017; Kim *et al.* 2017). Thus, while we have not done exhaustive model fitting due to computational constraints, our simulations reproduce estimates from studies which model a different facet of genetic variation (i.e. the site frequency spectrum).

Heterogeneous processes that correlate with a genomic feature will create differences in rates of substitution along the genome that correlate with the genomic feature. As shown in equation 1, this implies that the covariance along the genome between a genomic feature and divergence is expected to increase with time, and the rate of increase is equal to the covariance between that feature and the substitution rate. (It is important to note that varying covariances with ancestral diversity can be a confounding factor, and that the observation applies to covariance, not correlation.) Indeed, the covariance between divergence and recombination rate increases roughly linearly with time (see Fig. 10c), as expected because the rate of gBGC-induced fixations are correlated with recombination rate. Once this effect is removed (see Supplementary Fig. 7f), the covariance between exon density and divergence decreases linearly with time, as we would expect due to the effects of negative selection directly removing deleterious mutations in or near exons. The magnitude of this slope might produce a quantitative estimate of the strength of this effect, although more work is needed to disentangle confounders. It is important to contrast this observation, which applies mostly to the direct effects of selection, to other observations which also include linked effects (as discussed in Phung *et al.* 2016).

Although simulations allow for more biological realism, we made assumptions to constrain the parameter space explored. Our simulations used randomly mating populations of constant size, as inferred in Prado-Martinez *et al.* (2013). These population sizes were inferred using neutral model, and so are likely affected by the effects of selection (Jensen *et al.* 2005). Mean levels of diversity and divergence in our neutral simulation match the data, but simulations with natural selection differ, at times substantially (Supplementary Fig. 9). On the other hand, our simulations with selection match the data more closely with respect to standard deviation in levels of diversity and divergence along genomes. However, inaccuracy of the demographic model should not affect any of our main observations, because the effects of demography on levels of variation along genomes are not shared across multiple species.

We chose exons to be the targets of selection in our simulations. Exons cover about 1% of the human genome, and in reality selection affects noncoding regions as well. However, a substantial portion of this selection affects *cis*-regulatory regions, whose density along the genome is well predicted by coding sequence itself. Furthermore, highly conserved noncoding sequences have long been identified and characterized as functional (Bejerano *et al.* 2004; Siepel *et al.* 2005; Katzman *et al.* 2007). In the great apes, noncoding diversity is correlated with recombination rate, pointing to the role of selection (Castellano *et al.* 2020). However, it would be circular to include conserved noncoding elements in our simulations because such elements are identified based in part on levels of divergence, which themselves depend on ancestral levels of genetic diversity. Because conserved noncoding elements generally occur close to coding regions of the genome

(at the 1 Mb scale, the correlation between density of exons and PhastCons elements is around 0.6), we might expect a more realistic model to have the same amount of selection (in terms of total influx of selected mutations), but spread out over a somewhat wider region of the genome since we have omitted such sites. Even without considering all potential targets of selection, patterns of genetic diversity in our simulations match the data well: we see a correlation between simulated and observed diversity of 0.45 for chimps (Supplementary Fig. 8), for example.

While it has long been recognized that genetic variation among species might be structured similarly due to shared targets of selection, our results demonstrate that these correlations contain important information about the processes at work that has yet to be utilized fully. Here we have used large-scale simulations to demonstrate the combination of forces required to pattern shared divergence and diversity as we observe it in nature. Indeed, our results show that a combination of negative and positive selection, GC-biased gene conversion and mutation rate variation all contribute in shaping genetic variation in the great apes. Although some processes are not necessarily needed to recapitulate the real data (e.g. mutation rate variation), positive selection seems to be the only force that can explain most of our observations. There is clearly a need for future analytical work that might describe expected correlations across the genome given variation in local mutation, recombination, and selection. Furthermore, statistical model fitting based on theory or simulation is clearly desirable, although our experience suggests that the latter approach would prove computationally expensive.

## Data availability

Great apes empirical data are from Prado-Martinez *et al.* (2013), and it can be found at https://eichlerlab.gs.washington.edu/greatape/data.html. Code used in analyses and simulations can be found at https://github.com/kr-colab/greatapes_sims. Supplemental material is available at GENETICS online.

## Acknowledgements

We thank the Kern-Ralph CoLab for their invaluable support and input. Comments from reviewers also significantly improved this version of our paper.

## Funding

This work was supported by National Institutes of Health (NIH) awards R01GM117241 and R01HG010774 to ADK.

## Conflicts of interest

The author(s) declare no conflicts of interest.

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
