## [Peer Review File · Genetics]

Shared evolutionary processes shape landscapes of genomic variation in the great apes

Murillo Rodrigues, Andrew Kern, and Peter Ralph

NOTE: The reviews and decision letters are unedited and appear as submitted by the reviewers.

In extremely rare instances and as determined by a Senior Editor or the EIC, portions of a review may be redacted. If a review is signed, the reviewer has agreed to no longer remain anonymous.

The review history appears in chronological order.

Review Timeline:

Submission Date:	2023-06-20
Editorial Decision:	2023-08-14
Resubmission Received:	2023-10-26
Accepted:	2024-01-03

August 14, 2023

GENETICS-2023-306278

Shared evolutionary processes shape landscapes of genomic variation in the great apes

Dear Dr. Rodrigues:

Three experts in the field have reviewed your manuscript, and I have read it as well. The paper tackles an important long-standing question with a broad set of data, a new scale of simulations, and methodological innovation. While your manuscript is not currently acceptable for publication in GENETICS, we would welcome a substantially revised manuscript. Both reviewers have comments and concerns to be addressed in a revised manuscript. You can read their reviews at the end of this email.

To improve the manuscript for further consideration, you will need to address the reviewers' comments. The comments regarding further consideration of mutation rate variation (both in conjunction with negative selection, and parametrizing using known human maps) are especially important to consider, as are the comments regarding constraint on non-coding variation, relating the work to previous studies, and improving the presentation of results/conclusions of the manuscript. We look forward to receiving your revised manuscript. Please let the editorial office know approximately how long you expect to need for revisions.

Upon resubmission, please include:

1. A clean version of your manuscript;
2. A marked version of your manuscript in which you highlight significant revisions carried out in response to the major points raised by the editor/reviewers (track changes is acceptable if preferred);
3. A detailed response to the editor's/reviewers' feedback and to the concerns listed above. Please reference line numbers in this response to aid the editor and reviewers.

Your paper will likely be sent back out for review.

Additionally, please ensure that your resubmission is formatted for GENETICS
<https://academic.oup.com/genetics/pages/general-instructions>

Follow this link to submit the revised manuscript: Link Not Available

Sincerely,

John Novembre
Associate Editor
GENETICS

Approved by:
David Begun
Senior Editor
GENETICS

Reviewer #1 (Comments for the Authors (Required)):

The aim of the paper is to describe correlations of genetic diversity and divergence along a chromosome within and between species of great apes, and to test whether (and how) observed correlations can be re-created with forward time simulations.

The authors first report how the landscapes of both diversity and divergence are closely mirroring each other across the great apes. The phrasing "remarkable" stood out, since the pattern has been shown in other close species comparisons previously.

Secondly, they test if correlations between genomic landscapes of different species can be explained by the phylogenetic distance. They conclude that the correlations are stronger than would arise from ancestral variation alone, and so look for other genomic features to explain the excess correlation.

Rather than doing correlations with genomic features directly (which they instead return to later), the authors chose to make simulations in SLiM to try to recreate the found pattern between correlations between species landscapes and split time. For the

simulations, I would like to have a justification (read reference) both for the mutation rate and the selection coefficients they explored.

The researchers were unable to recreate the strong correlations between species landscapes found in the empirical data using simulations without selection. In the case where they achieved high correlations, the pattern of decreasing correlations with time since split between species was not there.

Finally, simulations were run that included natural selection. Simply deleterious mutations were not enough to cause strong correlations between landscapes. Positive selection can, on the other hand, lead to moderate correlations all by itself, and when both kinds of selection are included in the simulations, the correlations between landscapes and split time shown with the empirical data starts to appear.

I was wondering if the shared mutational landscape that was used in the neutral simulations were also added to the simulations that included selection? I got confused since the paper specifies that beneficial and deleterious mutations only can appear within exon (are there still other mutations happening in the non-coding regions?).

After having stated that recombination rate and functional density within the simulations are not enough to explain the patterns observed, the authors still go on to describe the relationship between these features and diversity/divergence in their group of species. I think for this analysis the authors should consider calculating "neutral" divergence and diversity metrics, by excluding snps in exons and conserved non-coding regions, similar to Chase, Ellegren, and Mugal (Evolution, 2021), to somewhat control for the effects of selection.

The authors find that the strength of correlations between landscapes of divergence and genomic features increase with time since split. They analytically decompose the covariance of genetic divergence and genomic features and suggest that the observed pattern most likely is caused by a covariance between substitution rates and another genomic feature (recombination rate or exon density). They believe that the increased divergence at the end of chromosomes with time are caused by GC-biased gene conversion, and that the negative covariance between exon density and divergence is due to purifying selection acting in these regions.

I think it is an ambitious paper, to use whole-chromosome simulations for comparison to the empirical data of multiple species. I don't have any specific suggestions for improvement. The code and simulation data should be made available.

Minor things: The figure legends need to be updated. Figure 2 has the first sentence repeated and refer to left and right panels while the panels are stacked. Figure 3 legend does not stand on its own (e.g. the purple line with a star is not explained, only in the main text). When referring to the correlation between exon density and diversity, it is Figure 10D that should be referred to (currently it says 10C). In the first definition of divergence, D_{xy} , time is noted as capital T and substitution rate with small r. The reverse is done later in the paper, and can be a source of confusion.

Reviewer #2 (Comments for the Authors (Required)):

The manuscript by Rodrigues et al. investigates how patterns of genetic variation change along the genomes in multiple different primate species. Patterns of diversity and divergence seem to covary with each other across species, even those that have diverged many millions of years ago. This finding would likely surprise many researchers in the field. The authors then use forward SLiM simulations to investigate potential evolutionary mechanisms for this pattern. They conclude that many factors can contribute-from mutation rate variation along the genome, negative natural selection, and positive selection to generate the observed signal.

Overall, I really enjoyed the manuscript. It makes some important empirical findings. It uses very large-scale simulations with as realistic models to interpret and understand the empirical patterns. The manuscript also contains a number of theoretical innovations that may be of interest to the readers of Genetics, such as the theory of the covariance between divergence/diversity and genomic variables (equation 1) as well as the use of PCA to compare the many correlations from the data to those of the models (Figure 10).

While I am enthusiastic about this study, I have a number of ideas to improve it:

1) One of the biggest things that is missing in this manuscript is selection on noncoding mutations. In the human and primate genomes, ~5% of the genome appears to be conserved across species by using PhyloP, GERP, PhastCons, etc. While it's still not entirely clear exactly how many of these conserved sites are under selection or how many noncoding sites under selection are actually in these extreme regions (see the higher estimate of 10.7% of the human genome under negative selection in Christmas et al. (2023 Science)), it is clear that most of the base pairs in the genome that are under selection are likely not within the coding regions. As such, selection on noncoding regions may have a profound impact on the patterns of

polymorphism and divergence across the genome, especially if the noncoding sequences under selection are in approximately orthologous positions across species.

While I appreciate that the forward in time simulations are extremely resource and time intensive, the authors should explore a model with deleterious mutations acting on noncoding mutations as well as on coding mutations. They could then test whether such a model would increase the correlations, resulting in them being more similar to those in the empirical data.

2) Another piece missing is that the authors consider mutation rate variation along the genome in isolation of the other models of selection. They find that a model with a standard deviation of around 0.08 can explain the correlations of diversity remaining high with increasing phylogenetic distance, but not the negative correlation between divergence and phylogenetic distance. Could a model with mutation rate variation and negative selection acting on nonsynonymous mutations better match the both of these patterns? Again, I realize that these simulations are time-consuming, and I'm not proposing that the authors should examine all combinations of all models. Rather, I think a couple illustrative cases, like those I'm suggesting here, would further understanding and more confidently rule in/or out models without positive selection.

3) Results Section 3.2 comes a little bit out of the blue here. The authors delve into the theoretical details of computing dT . At this point, it's not entirely clear why we need dT or what it is being used for. I think more context should be given. Maybe some of this could be discussed in Methods?

4) Related to the section described in my previous comment, how do the authors actually use the theory to compute dT from their data? Do they do this using the demographic model (including the split times) shown in Figure 1? Or is it informed by π or dXY ? More explanation here would help connect the model to what the authors are doing.

I just want to add, I think it's a really neat result that the correlations between diversity in different species decreases with phylogenetic distance between species, but still remains high even among distantly related species. This is a really important result!

I also have a few minor comments:

1) Line 46: The "," after "two species" is not needed.

2) Line 82: Linked selection can still impact divergence between species due to the polymorphism in the ancestral population. This was alluded to in the original Birky & Walsh paper cited here. It has been discussed further in Begun et al. (2007, see Figure 4) and Phung et al. 2016. Maybe mention this here.

3) Methods 2.2: Was any scaling of population size done in the SLiM simulations? It would be good to mention the population sizes that were used in the simulations.

4) Line 385: Maybe provide a reference for double-strand breaks being more common at the ends of chromosomes.

5) Figures 6 and 8: Maybe highlight on the figure which panels are qualitatively similar to the empirical correlations? That might make it easier for the readers to compare the models to the empirical correlations.

6) Line 463-469: Could this pattern be caused by larger species having larger neutral values of π . Then, there is a greater range for π to be reduced by selection. Put another way, in small populations with little genetic variation, there is little variation for selection to decrease, thus, not leaving much of a signal.

Reviewer #3 (Comments for the Authors (Required)):

In this manuscript, the authors use previously published genome sequences across closely related great apes to investigate the correlation between diversity patterns across species and between diversity and divergence patterns as well, all at the 1 Mb scale. They then perform extensive simulations with the aim of distinguishing factors that can cause such correlations. They conclude that some effect of natural selection and its effects on diversity (linked selection) appears necessary to explain the empirical correlations of diversity and divergence and that the best fitting simulations also include 10% of positively selected amino acid substitutions.

I find the study interesting and I agree with the authors that closely related species with diversity and divergence data should be powerful to distinguish between the many different evolutionary forces that can cause correlation - here they are mutation rate heterogeneity, GC-biased gene conversion, linked selection, and effective population size. However, even with all this data and all the simulations I still think the results are very inconclusive and descriptive - it almost seems like the complexities overwhelmed the authors and they then resorted to reporting their results and left it up to the reader to make more

interpretations and think of how to test them. This is my main criticism of the study that I hope the authors will aim to improve, including making the main display items easier to interpret, the specifics of the simulations easier to replicate and some more considerations of the underlying stochastic variance in the diversity and divergence vectors that are being correlated,.

I also have some more specific comments

1. I was surprised that the results are not related to previous studies of similar (perhaps even the same) data by main David Castellano, that investigate related phenomena and also exploit the close relationship of the species, two reference es below

(Impact of mutation rate and selection at linked sites on DNA variation across the genomes of humans and other homininaeD Castellano, A Eyre-Walker, K Munch

Genome Biology and Evolution 12 2020

Comparison of the full distribution of fitness effects of new amino acid mutations across great apesD Castellano, MC Macià, P Tataru, T Bataillon, K Munch

Genetics 213 (3), 953-966)

2. Why 1 MB? It seems to me that genetic drift may have a huge influence on the diversity data in species with small effective population size at this scale? How does that affect correlations? This could be more directly measured from the simulations. Could other scales, like 100Kb or 5 Mb be added for at least some of the analysis, the simulations would presumably not have to be redone.

3. In relation to this, how statistically significant is the claim the the "smile" in diversity/divergence is strongest for divergence - perhaps this is just from visual inspection? If that is the case, then I can see much more variation in diversity but also a smile if I imagine some smoothing. This is an important point to investigate before looking for reasons for a stronger smile in divergence.

4. At the 1Mb scale we already have data to estimate the heterogeneity in mutation rate, e.g. from the large decode study (Haldorsson et al 2019) with 3000 trios and almost 1 million de novos. I would really like the authors to use this source in order to at least constrain one of the many otherwise free parameters in their simulations.

5. Does the simulation results fit the average amount of diversity and divergence on each branch well and do the amount of ILS in the simulation fit published estimates of these? This is important to make a stronger test for neutrality alone. As I read it only one set of population sizes on each branch is being used but I will have to eyeball these from the Figure and I am not sure if they are reasonable. With selection included diversity is lost and the effective size in the simulation (ie the inverse of the average coalescence rate) will be smaller than the values simulated which presumably should be chosen from values from the empirical data. This possible circularity should be addressed.

6. I do not understand the PCA very well, how can the first PC explain 88% of the variance and PC2 only 8%. Is this correct? If so would it not mean that using the plot to display the difference between simulations and empirical values can be misleading since it is really their position along the X-axis that matters?

Response to Editor and Reviewers (GENETICS-2023-306278)

Associate Editor

Comment

To improve the manuscript for further consideration, you will need to address the reviewers' comments. The comments regarding further consideration of mutation rate variation (both in conjunction with negative selection, and parametrizing using known human maps) are especially important to consider, as are the comments regarding constraint on non-coding variation, relating the work to previous studies, and improving the presentation of results/conclusions of the manuscript. We look forward to receiving your revised manuscript. Please let the editorial office know approximately how long you expect to need for revisions.

Reply:

We thank the editor and reviewers for helpful feedback and respond in detail to their points below. In summary, we added new simulations of mutation rate variation with selection, expanded our analyses with new window sizes (now we have 500Kb, 1Mb and 5Mb), and polished our discussion to include relevant literature and a more detailed conclusion.

"consideration of mutation rate variation": Although we agree that parametrizing simulations using known human mutation rate variation maps would help determine the contribution of mutation rate variation to the patterns we uncovered, such a map does not exist yet. Reviewer 3 kindly pointed us to a paper with de-novo mutation rate calls. We explored the available data and we do not see how a proper mutation rate map can be estimated (at least without enough work to warrant another paper) because we have no information about variation in callability along the genome, and furthermore there are only 200,000 DNMs scattered along 3Gb of genome. Nevertheless, we computed the empirical standard deviation in mutation rate at the 1Mb scale (ignoring differences in callability and sampling noise) for the Haldorsson *et al.* (2019) dataset and found it is around 22%, which is inline with a previous paper that we include in our discussion (Smith *et al.*, 2018). We believe that building a mutation rate map for humans is beyond the scope of this paper, as it would entail gathering data from multiple papers (and much of that data is not freely available), and processing and developing models to control for batch and technology specific effects.

"constraint on non-coding variation": We agree that selection is not restricted to coding regions in natural populations. However, we limited our analyses to include selection in exons only for a few different reasons:

1. Many papers have produced maps of non-coding elements under constraint. Such maps are based on genetic divergence data, however. Because we are interested in figuring out the processes that affect divergence, it would be at least somewhat circular to include these in our analyses.

2. We restricted our analyses to large-scale patterns (i.e., mostly we focus on landscapes of diversity and divergence at the 1Mb scale). At this scale, we expect for *a priori* biological reasons that the density of exons to be a good stand-in for the density of targets of selection (e.g., promoters are adjacent to coding regions). Indeed, we found that PhastCons tracks and exon density are well correlated (~0.6). Thus, much of the selection on the non-coding elements should be picked up by neighboring exons. To demonstrate the adequacy of our simulations we have added two new features to the manuscript including: a new figure exploring the correlation in levels of diversity in the real data and in some of our simulations and additional text in the Discussion. Reassuringly, the correlation between levels of diversity in central chimps in the data and in our best fitting simulation is around ~0.45, which helps corroborate our assumption that exons carry enough information about the variation in the effects of selection along the genome of great apes.
3. Including selection on non-coding elements would expand even further the parameter space we would have to explore, but we are constrained by computational time and resources. Most of our simulations that include negative selection take over 30 days to run, despite our best efforts to parallelize independent branches in the phylogeny. The simulations are also RAM intensive, with usage surpassing 160Gb in many cases. Adding even more small regions under constraint would make simulations even more expensive.

Reviewer #1

Comment

The aim of the paper is to describe correlations of genetic diversity and divergence along a chromosome within and between species of great apes, and to test whether (and how) observed correlations can be re-created with forward time simulations. The authors first report how the landscapes of both diversity and divergence are closely mirroring each other across the great apes. The phrasing "remarkable" stood out, since the pattern has been shown in other close species comparisons previously. Secondly, they test if correlations between genomic landscapes of different species can be explained by the phylogenetic distance. They conclude that the correlations are stronger than would arise from ancestral variation alone, and so look for other genomic features to explain the excess correlation.

Reply:

We would like to thank Reviewer 1 for their careful reading of our paper! Although the observation is not completely novel, we believe it is still "remarkable". Furthermore, the great apes diverged over ~60N generations ago, which is longer than other taxa where similar patterns have been previously found, so we do not think the wording is exaggerated.

Comment

Rather than doing correlations with genomic features directly (which they instead return to later), the authors chose to make simulations in SLiM to try to recreate the found pattern between correlations between species landscapes and split time. For the simulations, I would like to have a justification (read reference) both for the mutation rate and the selection coefficients they explored.

Reply:

Thank you for catching this! We now added references for both these parameters. See lines 223-225.

Comment

The researchers were unable to recreate the strong correlations between species landscapes found in the empirical data using simulations without selection. In the case where they achieved high correlations, the pattern of decreasing correlations with time since split between species was not there. Finally, simulations were run that included natural selection. Simply deleterious mutations were not enough to cause strong correlations between landscapes. Positive selection can, on the other hand, lead to moderate correlations all by itself, and when both kinds of selection are included in the simulations, the correlations between landscapes and split time shown with the empirical data starts to appear. I was wondering if the shared mutational landscape that was used in the neutral simulations were also added to the simulations that included selection? I got confused since the paper specifies that beneficial and deleterious mutations only can appear within exon (are there still other mutations happening in the non-coding regions?).

Reply:

We now included simulations with selection and mutation rate variation. For discussion about the choice to include selection only within exons (while neutral mutations can happen anywhere in the genome), see the response to the Associate Editor above. See lines 443-462 and Fig. 8.

Comment

After having stated that recombination rate and functional density within the simulations are not enough to explain the patterns observed, the authors still go on to describe the relationship between these features and diversity/divergence in their group of species. I think for this analysis the authors should consider calculating "neutral" divergence and diversity metrics, by excluding snps in exons and conserved non-coding regions, similar to Chase, Ellegren, and Mugal (Evolution, 2021), to somewhat control for the effects of selection. The authors find that the strength of correlations between landscapes of divergence and genomic features increase with time since split. They analytically decompose the covariance of genetic divergence and genomic features and suggest that the observed pattern most likely is caused by a covariance between substitution rates and another genomic feature (recombination rate or exon density). They

believe that the increased divergence at the end of chromosomes with time are caused by GC-biased gene conversion, and that the negative covariance between exon density and divergence is due to purifying selection acting in these regions.

Reply:

Looking at some proxy for "neutral divergence" is an interesting idea, but we think that it would add additional complexity while being somewhat tangential to the main point of the paper. Our main aim was to understand the contribution of the direct effects of selection on the covariances between divergence and genomic features. That is, we are interested in figuring out what causes the changes in the strength of covariances over time. In the theory we develop, we argue that selection must be directly impacting substitution rates. Selection does not affect substitution rate of linked, neutral mutations, however (Birky and Walsh, 1988). Thus, if we were to use only "neutral" diversity and divergence, we would be removing the effects we are most interested in. In the Discussion, we point to past literature that studied the relationship between silent diversity and genomic features in the great apes (Nam et al., 2017).

See lines 494-496, 642-644.

Comment

*I think it is an ambitious paper, to use whole-chromosome simulations for comparison to the empirical data of multiple species. I don't have any specific suggestions for improvement. **The code and simulation data should be made available.***

Reply:

We have included a link to the GitHub repository with all the code used in this project. Although we are in favor of making all the simulation data available, we believe this is not feasible. We included 56 simulations of the entire great apes evolutionary history in the paper. Each resulting tree sequence is over 30Gb, so to make all the simulation data available we would need to host around 1.7Tb of data – Zenodo allows for 50Gb per project. We are not aware of any reasonable solutions to make such a large amount of data freely available, unfortunately. However, we are exploring whether making one or two simulations (e.g., the best-fitting simulation) would be useful and/or feasible.

See lines 238-239.

Comment

Minor things: The figure legends need to be updated. Figure 2 has the first sentence repeated and refer to left and right panels while the panels are stacked. Figure 3 legend does not stand on its own (e.g. the purple line with a star is not explained, only in the main text). When referring to the correlation between exon density and diversity, it is Figure 10D that should be referred to (currently it says 10C). In the first definition of divergence, D_{xy} , time is noted as capital T and substitution rate with small r. The reverse is done later in the paper, and can be a source of confusion.

Reply:

Thanks for catching these mistakes and typos! With respect to the definition of divergence in the beginning versus in the subsection *Correlations between genomic features and diversity and divergence*, we note that this distinction has a purpose. T is noted as capital in much of the literature when we refer to split times. In the next subsection however, we change the capitalization to reflect differences between scalars (not capitalized) versus vectors (capitalized).

Reviewer #2

Comment

The manuscript by Rodrigues et al. investigates how patterns of genetic variation change along the genomes in multiple different primate species. Patterns of diversity and divergence seem to covary with each other across species, even those that have diverged many millions of years ago. This finding would likely surprise many researchers in the field. The authors then use forward SliM simulations to investigate potential evolutionary mechanisms for this pattern. They conclude that many factors can contribute-from mutation rate variation along the genome, negative natural selection, and positive selection to generate the observed signal.

Overall, I really enjoyed the manuscript. It makes some important empirical findings. It uses very large-scale simulations with as realistic models to interpret and understand the empirical patterns. The manuscript also contains a number of theoretical innovations that may be of interest to the readers of Genetics, such as the theory of the covariance between divergence/diversity and genomic variables (equation 1) as well as the use of PCA to compare the many correlations from the data to those of the models (Figure 10).

While I am enthusiastic about this study, I have a number of ideas to improve it:

Reply:

We are happy that the reviewer is enthusiastic about the study and thankful that they provided detailed comments to further improve our paper.

1) One of the biggest things that is missing in this manuscript is selection on noncoding mutations. In the human and primate genomes, ~5% of the genome appears to be conserved across species by using PhyloP, GERP, PhastCons, etc. While it's still not entirely clear exactly how many of these conserved sites are under selection or how many noncoding sites under selection are actually in these extreme regions (see the higher estimate of 10.7% of the human genome under negative selection in Christmas et al. (2023 Science)), it is clear that most of the base pairs in the genome that are under selection are likely not within the coding regions. As such, selection on noncoding regions may have a profound impact on the patterns of

polymorphism and divergence across the genome, especially if the noncoding sequences under selection are in approximately orthologous positions across species.

While I appreciate that the forward in time simulations are extremely resource and time intensive, the authors should explore a model with deleterious mutations acting on noncoding mutations as well as on coding mutations. They could then test whether such a model would increase the correlations, resulting in them being more similar to those in the empirical data.

Reply:

We completely agree with the point that many of the targets of selection are not within the coding regions, however, in short think that our modeling choice is appropriate for the aims of this paper. See further discussion of this above in the response to the Associate Editor.

See lines 688-704.

2) Another piece missing is that the authors consider mutation rate variation along the genome in isolation of the other models of selection. They find that a model with a standard deviation of around 0.08 can explain the correlations of diversity remaining high with increasing phylogenetic distance, but not the negative correlation between divergence and phylogenetic distance. Could a model with mutation rate variation and negative selection acting on nonsynonymous mutations better match the both of these patterns? Again, I realize that these simulations are time-consuming, and I'm not proposing that the authors should examine all combinations of all models. Rather, I think a couple illustrative cases, like those I'm suggesting here, would further understanding and more confidently rule in/or out models without positive selection.

Reply:

This is a good suggestion; we have included new simulations where we added mutation rate variation on top of two of our simulations with selection. We found that with selection+mutation rate variation, you would need a bit less variance in mutation rates to get a better match to the data (compared to neutral+mutation rate variation simulations). Nevertheless, we found that a particular feature of the data – the fact that correlations decay when involving diversities computed in species with recent bottlenecks – can only be recovered when we add positive selection. This can be seen in Figures 4 and 7. This makes sense - consider that the strong bottleneck in humans won't affect the signatures of mutation rate variation along the genome much, and is expected to amplify the effects of BGS, but on the other hand, could interact with positive selection by erasing the effects of sweeps. Thus, if sweeps are causing correlations, we expect it to be more sensitive to the strong bottleneck in humans than the other processes.

See lines 443-462, 634-644 and Fig. 8.

3) Results Section 3.2 comes a little bit out of the blue here. The authors delve into the theoretical details of computing dT . At this point, it's not entirely clear why we need dT or what it

is being used for. I think more context should be given. Maybe some of this could be discussed in Methods?

Reply:

We have edited this section for clarity . Note that we have one section in the Methods discussing this as well.

See lines 248-265, 310-330.

4) Related to the section described in my previous comment, how do the authors actually use the theory to compute dT from their data? Do they do this using the demographic model (including the split times) shown in Figure 1? Or is it informed by pi or dXY? More explanation here would help connect the model to what the authors are doing.

Reply:

That's right (and, good question): dT is computed using the demographic model of Padro-Martinez et al (2013). This is now (hopefully) more clear in the paper.

See lines 252-253.

I just want to add, I think it's a really neat result that the correlations between diversity in different species decreases with phylogenetic distance between species, but still remains high even among distantly related species. This is a really important result!

Reply:

Thank you!

I also have a few minor comments:

1) Line 46: The "," after "two species" is not needed.

Reply: We could not find this in the text.

2) Line 82: Linked selection can still impact divergence between species due to the polymorphism in the ancestral population. This was alluded to in the original Birky & Walsh paper cited here. It has been discussed further in Begun et al. (2007, see Figure 4) and Phung et al. 2016. Maybe mention this here.

Reply: Good point! We now mention this effect in the main text. **See lines 82-83.**

3) Methods 2.2: Was any scaling of population size done in the SLiM simulations? It would be good to mention the population sizes that were used in the simulations.

Reply: No scaling was done. In the Methods we now highlight that the population sizes are the same as shown in Fig. 1. **See lines 202-203.**

4) Line 385: Maybe provide a reference for double-strand breaks being more common at the ends of chromosomes.

Reply: We added a citation. **See line 387.**

5) Figures 6 and 8: Maybe highlight on the figure which panels are qualitatively similar to the empirical correlations? That might make it easier for the readers to compare the models to the empirical correlations.

Reply: We agree that might make it easier for readers, but worry that there is already a lot of information in those panels. The comparison should be easier to see in our PCA plots.

Nevertheless, we highlighted simulations closer to the data in the captions of Figures 7 and 8.

6) Line 463-469: Could this pattern be caused by larger species having larger neutral values of π . Then, there is a greater range for π to be reduced by selection. Put another way, in small populations with little genetic variation, there is little variation for selection to decrease, thus, not leaving much of a signal.

Reply: We are not sure what the reviewer means by this. Because we use correlations (and not covariances), differences in standard deviation should be accounted for.

Reviewer #3

In this manuscript, the authors use previously published genome sequences across closely related great apes to investigate the correlation between diversity patterns across species and between diversity and divergence patterns as well, all at the 1 Mb scale. They then perform extensive simulations with the aim of distinguishing factors that can cause such correlations. They conclude that some effect of natural selection and its effects on diversity (linked selection) appears necessary to explain the empirical correlations of diversity and divergence and that the best fitting simulations also include 10% of positively selected amino acid substitutions. I find the study interesting and I agree with the authors that closely related species with diversity and divergence data should be powerful to distinguish between the many different evolutionary forces that can cause correlation - here they are mutation rate heterogeneity, GC-biased gene conversion, linked selection, and effective population size. However, even with all this data and all the simulations I still think the results are very inconclusive and descriptive - it almost seems like the complexities overwhelmed the authors and they then resorted to reporting their results and left it up to the reader to make more interpretations and think of how to test them. This is my main criticism of the study that I hope the authors will aim to improve, including making the main display items easier to interpret, the specifics of the simulations easier to replicate and some more considerations of the underlying stochastic variance in the diversity and divergence vectors that are being correlated.

Reply:

This is a fair criticism - we are aiming to write a paper describing this interesting data, and could not definitively answer all the (very big!) questions that we would like to. To address this, we have tried in the revisions to make it more explicit what we think we can confidently conclude and what is still uncertain. In particular, we polished our discussion to make our conclusion clearer and to delineate what we think are the next steps in this area of research.

See lines 634-644, 705-718.

I also have some more specific comments

1. *I was surprised that the results are not related to previous studies of similar (perhaps even the same) data by main David Castellano, that investigate related phenomena and also exploit the close relationship of the species, two reference es below*

(Impact of mutation rate and selection at linked sites on DNA variation across the genomes of humans and other homininaeD Castellano, A Eyre-Walker, K Munch

Genome Biology and Evolution 12 2020

Comparison of the full distribution of fitness effects of new amino acid mutations across great apesD Castellano, MC Macià, P Tataru, T Bataillon, K Munch

Genetics 213 (3), 953-966)

Reply:

Thank you for catching this major oversight! We meant to cite these papers, as they have made important contributions to understanding the role of different evolutionary processes in shaping variation in the great apes. We have now included the appropriate citations and discuss some of their results.

See lines 224, 619, 693-694.

2. *Why 1 MB? It seems to me that genetic drift may have a huge influence on the diversity data in species with small effective population size at this scale? How does that affect correlations? This could be more directly measured from the simulations. Could other scales, like 100Kb or 5 Mb be added for at least some of the analysis, the simulations would presumably not have to be redone.*

Reply:

We now repeated our analyses at 500Kb and 5Mb. We found that qualitatively the results remain the same. Part of the reason why we did not include smaller scales was because our recombination map was at the 1Mb resolution. Indeed, we see that at 500Kb the data is slightly further away from our simulations. On the other hand, analysis at larger scales suffer because of sampling noise in estimating correlations (we can divide chr12 in 26 5Mb windows only). We see that at 5Mb the results largely agree with 1Mb.

See lines 476-481 and Figures S5, S6.

3. *In relation to this, how statistically significant is the claim that the "smile" in diversity/divergence is strongest for divergence - perhaps this is just from visual inspection? If that is the case, then I can see much more variation in diversity but also a smile if I imagine some smoothing. This is an important point to investigate before looking for reasons for a stronger smile in divergence.*

Reply:

This is a qualitative remark from visual inspection, however we are confident that the result would be strongly statistically significant using an appropriate statistical test. However, the exact test to use is not obvious to us, and we don't think it would add to the manuscript sufficiently to figure out how to do the test. (And, to be clear: the claim is not that there is no smile in diversity, but that it becomes more pronounced over time due to divergence accumulating faster on the ends of chromosomes.) We think that it might be more plausible to the reviewer from looking at Figure S4, which shows (we think clearly?) that divergence accumulates faster in the ends of the chromosome only in certain classes of sites, i.e., that the "smile" goes away if we condition on sites which were "strong" in the ancestor of the great apes (and which are not affected by gBGC). The pattern is also fairly (we think) unambiguous in Fig. S1, which shows that divergence increases faster with time in the high recombination regions. Note this has been observed before, and we mention this has been seen in birds (Ellegren et al. 2012) in the Discussion.

See lines 603-609.

4. At the 1Mb scale we already have data to estimate the heterogeneity in mutation rate, e.g. from the large decode study (Haldorsson et al 2019) with 3000 trios and almost 1 million de novos. I would really like the authors to use this source in order to at least constrain one of the many otherwise free parameters in their simulations.

Reply:

We thank the reviewer for raising this point that a mutation map is necessary for better understanding what shapes variation in humans and other great apes. As discussed in more detail in the response to the Associate Editor, Haldorsson et al. (2019) identified 200,000 de novo mutations (DNMs), which, compounded with the fact that callability is not available, does not seem sufficient for building a good mutation rate map. There are more DNM studies out there, but collating all these datasets is beyond the scope of this paper.

5. Does the simulation results fit the average amount of diversity and divergence on each branch well and do the amount of ILS in the simulation fit published estimates of these? This is important to make a stronger test for neutrality alone. As I read it only one set of population sizes on each branch is being used but I will have to eyeball these from the Figure and I am not sure if they are reasonable. With selection included diversity is lost and the effective size in the simulation (ie the inverse of the average coalescence rate) will be smaller than the values simulated which presumably should be chosen from values from the empirical data. This possible circularity should be addressed.

Reply:

This is an important point, and a problem in much of population genetics. Ideally, one would like to co-infer demography and selection parameters, since estimated effective population sizes are in practice affected by selection. We included a new figure showing how our simulations compare to our data, as the reviewer requests, in basic summaries. We find that indeed the

simulations with selection have lower overall diversity than the data, probably due to the fact that the demographic inference did not take selection into account. However, our simulations with selection are closer to the data with respect to variance in levels of diversity along chromosomes. It is beyond the scope of this paper to co-infer demography and selection, however. See the new paragraph in the Discussion.

See lines 678-687.

6. I do not understand the PCA very well, how can the first PC explain 88% of the variance and PC2 only 8%. Is this correct? If so, would it not mean that using the plot to display the difference between simulations and empirical values can be misleading since it is really their position along the X-axis that matters?

Reply:

We have more simulations with mutation rate variation than with different parameters for selection. These differences dominate PC1 and make it explain more variance. The differences captured in PC2 are not necessarily less important in determining the simulation that best fits the data. Now that we added new simulations with both mutation rate variation and selection, it becomes clearer that the simulations without positive selection do not fit the data as well. But our results do not discard the possibility that mutation rate variation is an important driver of genetic variation in the great apes. The (good) question that the reviewer raises is analogous to the question of “how many PCs should we use?” We think that the answer to this - at least in this case - is that if the answer depends critically on the choice of number of PCs, then we should distrust the results. Happily, we can see that it does not: from Figure 9, the closest simulations in PC1+PC2 space (as drawn in that Figure) only differ in minor aspects from the simulations that are closest using PC1 only.

See lines 472-475.

January 3, 2024

RE: GENETICS-2023-306567

Dr. Murillo F. Rodrigues
University of Oregon
Department of Biology
5289 University of Oregon
Institute of Ecology and Evolution
Eugene, Oregon

Dear Dr. Rodrigues:

Congratulations! We are delighted to inform you that your manuscript entitled "Shared evolutionary processes shape landscapes of genomic variation in the great apes" is acceptable for publication in GENETICS. Many thanks for submitting your research to the journal.

The reviewers had a few suggestions for improving the manuscript that you may want to consider. You can view their comments at the bottom of this email.

To Proceed to Production:

1. Format your article according to GENETICS style, as discussed at <https://academic.oup.com/genetics/pages/general-instructions>, and upload your final files at <https://genetics.msubmit.net>.
2. Your manuscript will be published as-is (unedited-as submitted, reviewed, and accepted) at the GENETICS website as an Advanced Access article and deposited into PubMed shortly after receipt of source files and the completed license to publish. Please notify sourcefiles@thegsajournals.org if you do not wish to publish your article via Advanced Access.
3. We invite you to submit an original color figure related to your paper for consideration as cover art. Please email your submission to the editorial office or upload it with your final files. You can submit a small-sized image for evaluation, and if selected, the final image must be a TIFF file 2513px wide by 3263px high (8.375 by 10.875 inches; resolution of 600ppi). Please avoid graphs and small type.

If you have any questions or encounter any problems while uploading your accepted manuscript files, please email the editorial office at sourcefiles@thegsajournals.org.

Also, we appreciate your patience with the decision during a busy December, and hope this is excellent news to start your year. Thanks again for sending the work to Genetics and for developing this nice contribution,

Sincerely,

John Novembre
Associate Editor
GENETICS

Approved by:
David Begun
Senior Editor
GENETICS

note: Please add jnls.author.support@oup.com and genetics.oup@kwglobal.com (or the domains @oup.com and @kwglobal.com) to your email program's "safe senders" list. You will be contacted by both at various points during the production process.

Review comments (if applicable):

Reviewer #2 (Comments for the Authors (Required)):

The authors have addressed some of my concerns and comments on the previous version of the manuscript. In particular, I appreciated the consideration of models with selection and mutation rate variation. I wish the authors had considered the role of selection noncoding variation more fully, but I see their point that such simulations would be very computationally demanding and we do not have a good idea of the parameters for these simulations anyway. Given the overall importance of the empirical findings, I think it is fine to leave the noncoding simulations for future work.

I only have a few very minor comments on the revision:

1) Kong et al. 2002 and Kong et al. 2010 are both cited for the recombination maps. These are different papers and different maps. Please double-check which one was used and cite the appropriate reference.

2) Line 241: "neural" should be "neutral".

Reviewer #3 (Comments for the Authors (Required)):

The authors have addressed all the comments from me and the other reviewers, at least in the rebuttal.

I think it is understandable that they leave some of the suggestions for future work. The rough considerations of empirical mutation rate variation is sufficient for the main arguments in my opinion and I am happy to see it included.

I am still a bit dissatisfied with two particular points, namely: 1. the variances in the PCAs and their interpretations, it is hard not to read the distances on the two axes as being equally important which they clearly are not, and 2: the differences in "smiles" in divergence and diversity. I understand that making a proper statistical test of these claims is beyond the scope, but I also think that eyeballing is a tricky thing here since the scales on the Y-axis are so different. I would try to make it scale invariant by just plotting the proportion of the max for each curve and smoothing the diversity plots to have similar variances along the chromosomes as the divergence plots. However, these are just suggestions to consider once more.